

# A hydrodynamic model for Galveston Bay and the shelf in the northwestern Gulf of Mexico

Jiabi Du[1], Kyeong Park[1], Jian Shen[2], Yinglong J. Zhang[2], Xin Yu[2], Fei Ye[2], Zhengui Wang[3], Nancy N. Rabalais[4]

[1]Department of Marine Sciences, Texas A&M University at Galveston, Galveston, TX 7754, USA

[2]Virginia Institute of Marine Science, College of William and Mary, Gloucester, VA 23062, USA

[3]School of Marine Sciences, University of Maine, Orono, ME 04469, USA

[4]Louisiana State University, Baton Rouge, LA 70803, USA

*Correspondence to*: Jiabi Du (jdu@tamug.edu)

**Abstract:** We present a 3D unstructured-grid hydrodynamic model for the northwestern Gulf of Mexico that utilizes a high-resolution grid for the main estuarine systems along the Texas-Louisiana coast. This model, based on the Semi-implicit Cross-scale Hydroscience Integrated System Model (SCHISM) with hybrid horizontal and vertical grids, is driven by the observed river discharge, reanalysis atmospheric forcing, and open boundary conditions from the global models. The model reproduces well the temporal and spatial variation of observed water level, salinity, temperature, and current velocity both in Galveston Bay and on the shelf. We apply the validated model to examine the remote influence from large rivers, specifically the Mississippi and Atchafalaya rivers, on the salinity regime along the Texas coast. Numerical experiments reveal that the Mississippi-Atchafalaya discharge could significantly decrease the salinity on the inner shelf along the Texas coast and its influence highly depends on the wind field and the resulting shelf current. Winter wind tends to constrain the Mississippi-Atchafalaya water against the shore, forming a narrow lower-salinity band all the way to the southwestern Texas coast. Under summer wind, the influence of the discharge on salinity is limited to the upper Texas coast while extended offshore. The decrease in salinity at the mouth of Galveston Bay due to the Mississippi-Atchafalaya discharge leads to a decrease in horizontal density gradient, a weakened estuarine circulation inside the bay, a decrease in the salt flux, and a smaller estuarine-ocean exchange. We highlight the flexibility of the model that simulates not only estuarine dynamics and shelf-wide transport but also the interaction between them.



## 1. Introduction

Northern Gulf of Mexico (GoM) is characterized by complicated shelf and coastal processes including multiple river plumes with varying spatial scales, highly energetic deep-current due to steep slopes, upwelling in response to alongshore wind forcing, and mesoscale eddies derived from Loop Currents of Gulf Stream (Oey et al., 2005; Dukhovskoy et al., 2009; Dzwonkowski et al., 2015; Barkan et al., 2017). Freshwater derived from the Mississippi-Atchafalaya River basin introduces excess nutrients and terminates amidst one of the United

States' most productive fishery regions and the location of the largest zone of hypoxia in the western Atlantic Ocean (Rabalais et al., 1996, 2002; Bianchi et al., 2010). The physical, biological, and ecological processes have been attracting increasing attention, given the region's sensitive response to large-scale climate variation, accelerated sea-level rise, and strong anthropogenic interventions (Justić et al., 1996; Rabalais et al. 2007).

        Understanding the interaction and coupling between regional scale ocean dynamics and local scale

estuarine processes is of great interest. Many observational (in-situ/satellite) (e.g., Cochrane and Kelly, 1986; DiMarco et al., 2000; Chu et al., 2005) and numerical modeling (structured/unstructured grid) (Zavala-Hidalgo et al., 2003, 2006; Hetland and Dimarco, 2008; Fennel et al., 2011; Gierach et al., 2013; Huang et al., 2013) studies have been conducted for the GoM shelf. Hetland and Dimarco (2008) configured a hydrodynamic model based on the Regional Ocean Modelling System (ROMS: Shchepetkin and McWilliams, 2005) for the Texas-

Louisiana shelf, which has been used for the subsequent physical and/or biological studies (Fennel et al., 2011; Laurent et al., 2012; Rong et al., 2014). Zhang et al. (2012) extended the model domain westward to cover the entire Texas coast. Wang and Justić (2009) implemented the Finite Volume Coast Ocean Model (FVCOM: Chen et al., 2006) over the similar domain of Hetland and Dimarco (2008). Lehrter et al (2013) implemented the Navy Coastal Ocean Model (NCOM: Martin, 2000) over the inner Louisiana shelf with focus on the

Mississippi River plumes. In addition, there were modeling studies for larger domains such as the entire GoM (Oey and Lee, 2002; Wang et al., 2003; Zavala-Hidalgo et al., 2003). For example, Zavala-Hidalgo (2003) using the NCOM investigated the seasonally varying shelf circulation in the western shelf of the GoM. Bracco et al. (2016) using the ROMS examined the mesoscale and sub-mesoscale circulation in the northern Gulf of Mexico.

Other hydrodynamic modeling studies focused on specific estuarine systems such as Galveston Bay (Rayson et al., 2015, 2016; Rego and Li, 2010; Sebastian et al., 2014), Mobile Bay (Kim and Park, 2012; Du et al., 2018a), and Choctawhatchee Bay (Kuitenbrouwer et al., 2018). These models focusing on a single estuarine system tend to have smaller domains, including the target estuary and the inner shelf just outside of the estuary and used the ocean boundary conditions based on data and/or global (e.g., HYCOM) or regional models. Many

coastal bays along the Texas-Louisiana coast are affected by the conditions on the shelf that is regulated by the



riverine buoyancy and materials from neighboring estuarine systems and the shelf transport processes. In addition, dynamics in an estuarine system are also altered by localized small-scale geometric and bathymetric features such as narrow but deep ship channels, seaward extending jetties, and offshore sandbars, which are typically on the order of 10 to 100 m. Coupling between the estuarine and shelf processes is therefore critical

for a more comprehensive understanding of the processes in both the shelf and estuaries in the northern GoM. With the importance of the capability of a model of an estuary to simulate the coupling, cross-scale models with unstructured grids become an attractive option.

The hydrodynamic conditions, particularly the salinity, over the Louisiana shelf is dominated by the influence of the Mississippi-Atchafalaya river plumes (Lehrter et al, 2013; Rong et al., 2014; Androulidakis et

al., 2015). However, their effect on the salinity on the Texas shelf has not been well documented. Measurements at Port Aransas (600 km to the west of Atchafalaya River) show evident seasonal cycle, with higher salinity during the summer (Bauer, 2002). Is this seasonality related to the seasonal variation of remote influence from the Mississippi-Atchafalaya rivers? A broader question may be what would be the effect of the Mississippi-Atchafalaya discharge on the salinity along the Texas coast. Furthermore, it is also important to understand the

temporal and spatial scales with which the salinity at or near the mouth of an estuarine bay system may respond to the river plumes from neighboring systems. For example, how long will it take for the salinity at the Texas coast to respond to a pulse of freshwater input from the Mississippi-Atchafalaya rivers? This time scale in comparison to the time scales of estuarine processes (e.g., recovery time scale from storm disturbance) will allow one to determine the importance of the remote influence from major rivers.

Here, we present a model for the northwestern GoM that includes all the major estuaries as well as the shelf in the model domain and construct a grid with fine resolution for local estuaries. Using Galveston Bay as an example, we demonstrate the importance of the interactions among estuaries and the shelf by investigating the remote influence from the Mississippi and Atchafalaya rivers on the salinity regime along the Texas coast.

## 2. Methodology

**2.1 Model description**

We employed the Semi-implicit Cross-scale Hydroscience Integrated System Model (SCHISM: Zhang et al., 2015, 2016), an open-source community-supported modeling system based on unstructured grids, derived from the early SELFE model (Zhang and Baptista, 2008). The model is based on the turbulence-averaged Navier-Stokes equations, including continuity, momentum, salt-balance, and heat-balance equations, under the

hydrostatic approximation. It uses a semi-implicit Galerkin finite-element method with an Eulerian-Lagrangian

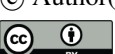



algorithm for momentum advection and a finite-volume method with an upwind or TVD$^2$ scheme for tracer

advection (Casulli and Zanolli, 2005; Ye et al., 2017). It uses the generic length-scale model of Umlauf and

Burchard (2003) with the stability function of Kantha and Clayson (1994) for turbulence closure. One of the

major advantages of the model is that it has the capability of employing a very flexible vertical grid system,

robustly and faithfully resolving the complex topography in estuarine and oceanic systems without any

smoothing (Zhang et al., 2016; Stanev et al., 2017; Du et al., 2018b; Ye et al., 2018). A more detailed

description of the SCHISM, including the governing equations, horizontal and vertical grids, numerical solution

methods, and boundary conditions, can be found in Zhang et al. (2015, 2016).

### 2.2 Model domain and grid system

The model domain covers the Texas, Louisiana, Mississippi, and Alabama coasts, including the shelf as

well as major estuaries (e.g., Mobile Bay, Mississippi River, Atchafalaya River, Sabine Lake, Galveston Bay,

Matagorda Bay, and Corpus Christi Bay) (Fig. 1). The domain also includes part of the deep ocean to set the

open boundary far away from the shelf so as to avoid imposing boundary conditions at topographically complex

locations. The grid system contains 142,972 horizontal elements (triangular and quadrangular), with the

resolution ranging from 10 km in the open ocean to 2.5 km on average on the shelf shallower than 200 m and 40

m in the narrow ship channel of Galveston Bay . The fine grid for the ship channel is carefully aligned with the

channel orientation in order to accurately simulate the salt intrusion process (Ye et al., 2018). Vertically, a

hybrid $s$-$z$ grid is used, with 10 sigma layers for waters less than 20 m and another 30 $z$ layers for depths from

20 to 4000 m (20, 25, 30, 35, 40, 50, 60, 70, 80, 90, 100, 125, 150, 200, 250, 300, 350, 400, 500, 600, 700, 800,

900, 1000, 1250, 1500, 2000, 2500, 3000, 4000 m); shaved cells are automatically added near the bottom in

order to faithfully represent the bathymetry and thus the bottom-controlled processes, e.g., dense water outflow.

This hybrid $s$-$z$ vertical grid enables the model to better capture the stratification in the upper surface layer while

keeping the computational cost reasonable for simulations of the deep waters. The computational efficiency of

this model set-up is very good. With a time step of 120 s and the TVD$^2$ scheme for mass transport, it takes about

24 hrs for one-year simulation with 120 processors (Intel Xeon E5-2640 v4).

The bathymetry used in the model is based on the coastal relief model (3 arc-second resolution:

https://www.ngdc.noaa.gov). The local bathymetry in Galveston Bay is augmented by 10-m resolution DEM

bathymetric data to resolve the narrow ship channel (150 m wide, 10-15 m deep) that extends from the bay

entrance all the way to Port of Houston. Bathymetry of the ship channels in other rivers, such as Mississippi,

Atchafalaya, and Sabine rivers, is manually set following the NOAA navigational charts. The depth in the

model domain ranges from 3400 m in the deep ocean to less than 1 m in Galveston Bay (Fig. 2).



### 2.3 Forcing conditions

The model was validated for the two-year conditions in 2007-2008 and was forced by the observed river

discharge, reanalysis atmospheric forcing, and open boundary conditions from the global models. Daily

freshwater inputs from the USGS gaging stations were specified at 15 river boundaries (Fig. 1). For the

Mississippi River, the largest in the study area, river discharge at Baton Rouge, LA (USGS 07374000) was

used. For the Atchafalaya River, the discharge data at the upper river station (USGS 07381490 at Simmesport,

LA) are not available before 2009. However, we found a significant linear relationship between this station and

the one near mouth (USGS 07381600 at Morgan City, LA) with $r^2$ of 0.92 with a 2-day time lag based on the

data from 2009 to 2017. The freshwater discharge estimated using this relationship was used to specify the

freshwater input at Simmesport discharging into the Atchafalaya Bay. For the Trinity River, the major river

input for Galveston Bay, river discharge at the lower reach station at Wallisville (USGS 08067252) was used,

where the mean river discharge (averaged over April 2014 and April 2018) is about 56% of that at an upper

reach station at Romayor (USGS 08066500). This is because the water from Romayor likely flows into

wetlands and water bodies surrounding the main channel of the Trinity River before reaching Wallisville

(Lucena and Lee, 2017). The river discharge data at the Wallisville station are not available before April 2014.

However, there is a significant linear relationship between these two stations ($r^2$ of 0.89 with a 4-day time lag

based on the data from 2014 to 2018). The freshwater discharge estimated using this relationship was used to

specify the freshwater input at Wallisville discharging into Galveston Bay. River flows from other rivers were

prescribed using the data at the closest USGS stations. Water temperatures at the river boundaries were also

based on the data at these USGS stations.

Reanalyzed 0.25° resolution, 6 hourly atmospheric forcing, including air temperature, solar radiation,

wind, humidity, and pressure at mean sea level, were extracted from the European Centre for Medium-Range

Weather Forecasts (ECMWF: https://www.ecmwf.int). Both harmonic tide and subtidal water level were used

to generate water level ocean boundary condition, with the harmonic tide (M2, S2, K2, N2, O1, Q1, K1, and P1)

extracted from the global tidal model FES2014 (Carrere et al., 2015) and the subtidal water level from the low-

pass filtered (cut-off period of 15 days) daily global HYCOM data. The model was relaxed during inflow to the

subtidal variation of HYCOM output at the open boundary in terms of salinity, temperature, and velocity.

### 2.4 Numerical experiments

To investigate the remote influence from the major river plumes from the Mississippi-Atchafalaya river

system, we conducted three idealized numerical experiments with constant long-term mean river discharges: (1)

the river discharges into Galveston Bay only with the January 2008 wind field (Jan-G); (2) the Mississippi-



Atchafalaya river discharge as well as the discharges into Galveston Bay with the January 2008 wind field (Jan-GAM); and (3) the same discharges as (2) but with the July 2008 wind field (Jul-GAM). To examine the effect

of different seasonal winds, we chose the January 2008 and July 2008 winds as a representative of the winter and summer winds, respectively. The January or July wind was repeated every month in the simulation to take into account the wind variability, which is known to play an important role on the shelf circulation (Ohlmann and Niiler, 2005). The January wind is dominated by northeasterly wind, while the July wind is dominated by southwesterly wind, which will result in distinctly different shelf currents, i.e., downcoast and upcoast shelf

currents under the January and July winds, respectively. The boundary conditions were based on the two-year mean conditions (temporally constant but spatially varying), with the exception of salinity set at a constant value of 36 psu along the open boundary. Starting with the spatially uniform constant salinity of 36 psu throughout the model domain, each model run shows the effect on salinity dilution under the specified condition of river discharge and wind.

**3. Model validation**

The model results of 2007-2008 were compared with observations for water level at seven NOAA tidal gauge stations, salinity at four Texas Water Development Board (TWDB) stations, temperature at three NOAA stations, and current velocity at two Texas Automated Buoy System (TABS) buoys (see Fig. 2 for station locations). Comparisons were made for both total and subtidal (48-hr low-pass filtered) components. For

quantitative assessment of the model performance, two indexes were used, model skill and mean absolute error (*MAE*) (Wilmott, 1981):

$$Skill = 1 - \frac{\sum_{i=1}^{N} |X_{\mathrm{mod}} - X_{obs}|^2}{\sum_{i=1}^{N} (|X_{\mathrm{mod}} - \overline{X_{obs}}| + |X_{obs} - \overline{X_{obs}}|)^2} \qquad (1)$$

$$MAE = \frac{1}{N} \sum_{i=1}^{N} |X_{\mathrm{mod}} - X_{obs}| \qquad (2)$$

where $X_{obs}$ and $X_{mod}$ are the observed and modeled variables, respectively, with the overbar indicating the

temporal average over the number of observations (*N*). *Skill* provides an index of model-data agreement, with a skill of one indicating perfect agreement and a skill of zero indicating complete disagreement. The magnitude of *MAE* indicates the average deviation between model and data.



### 3.1 Water level

The model-data comparisons were made for water level at stations along the coast and inside Galveston
Bay. Manning's friction coefficient was used as a calibration parameter. The model results with a spatially
uniform Manning's coefficient of 0.016, which is then converted to the bottom drag coefficient for the 3D
simulation, gives a good agreement between model and data. Overall, the model reproduces well both the tidal
and subtidal components of water level at tidal gauge stations along the coast as well as inside Galveston Bay
(Fig. 3 and Table 1). The *MAE* is in the range of 7-11 cm and 5-7 cm for the total and subtidal components,
respectively. The model skill varies spatially, with relatively low skills occurring at Pilot Station and Dauphin
Island, which might be due to their proximity to the open boundary and errors in the boundary conditions. Good
skills occur at the stations in the Texas coast including Galveston Bay, with the skills larger than 0.92 for the
total and subtidal components. It is interesting to note that the model has reproduces well the peak in water level
during Hurricane Ike (around day 625), one of the most severe hurricanes that hit the Houston-Galveston area in
recent years. Simulation of surface elevation is sensitive to topography, bottom friction, boundary conditions,
and atmospheric forcings. Some discrepancies are expected due to the assumption of spatially uniform bottom
friction coefficient. Further improvement might be achieved by using spatially varying coefficients, but we did
not deem it worth applying, considering the current performance of the model. Additional discrepancies may
come from the limited spatial and temporal resolution of atmospheric forcings and the reliability of the open
boundary conditions from the global HYCOM output, especially for stations close to the open boundary.

### 3.2 Salinity

The model reproduces well the pattern of variations in observed salinity at stations inside Galveston Bay
(Fig. 4 and Table 1). The *MAE* is no larger than 4 psu for the total and subtidal components, and the model skill
ranges from 0.76-0.93 and 0.69-0.93 for the total and subtidal components, respectively. It is important to note
that the salinity at the bay mouth under normal (e.g., non-flooding) condition is sensitive to the alongshore
transport of low salinity water from adjacent estuaries, such as nearby Sabine-Neches rivers, Atchafalaya River,
and Mississippi River. A good reproduction of salinity at the bay mouth requires the model performance for not
only the bay-wide transport but also the shelf transport. Errors in modeled salinity at the bay mouth can
propagate to the upper bay. For example, salinity during days 60-100 is overestimated at the mouth (station
BOLI) and this error propagated into the middle bay station (station MIDG) (Fig. 4). Discrepancies as large as
10 psu are not likely caused by inaccurate discharge from the Trinity River, as this river has a very limited
influence on the salinity on the shelf (further discussed in Section 4.2). After day 100, the model reproduces the



salinity at the mouth reasonably well. Good model skills at the mouth, again, suggests the good model

performance in simulating the low-salinity water transport along the shelf as well as the along-bay transport.

210        It is worth noting that the model reproduces well the sharp change of salinity during Hurricane Ike (Fig.

4). The salinity at the upper bay (BAYT) decreased from 26 to 0 psu within two days, which was caused by a

pulse of freshwater discharge from Lake Houston (see reservoir storage at USGS 08072000). In addition, the

model reproduces well the spatial difference in the amplitude of tidal fluctuation of salinity (Fig. 4). The salinity

in Trinity Bay (station TRIN) shows very weak tidal signal while that at the bay mouth (station BOLI) shows

much stronger tidal signal. Galveston Bay in general has micro-tidal ranges with a mean tidal range of 0.3 m at

a mid-bay station (Eagle Point in Fig. 2). The tidal signal, however, becomes stronger at the narrow bay mouth

(2.5 km wide), with the tidal current as strong as 1 m s$^{-1}$ (see station g06010 at

http://pong.tamu.edu/tabswebsite/)

        The modeled salinity was also compared to the observed salinity structure over the Texas-Louisiana

shelf using the data from the shelf-wide summer survey in July 2008 as an example (Fig. 5). Both the horizontal

and vertical structures of salinity on the shelf are well reproduced by the model, with the *MAE* over 65 stations

of 1 and 2 psu for the surface and bottom salinity, respectively. Both data and model show relatively shallow

halocline at section A (west of Mississippi Delta) and deeper halocline at section F (off Atchafalaya Bay). The

upper layer salinity off Atchafalaya Bay was nearly well mixed, which is very well reproduced by the model,

although the model somewhat underestimates the bottom salinity at section F. Note the small variability within

one day of the modeled vertical salinity profile on the shelf, e.g., stations F4 and A7 in Fig. 5.

### 3.3 Temperature

        The model has reproduced well the observed temperatures at three NOAA stations located from the

Galveston Bay mouth to the upper bay (Fig. 6). Both the seasonal and diurnal cycles are well captured, with the

*MAE* of 1°C and the model skills larger than 0.97. Even within a relatively small region inside Galveston Bay,

temperature can vary spatially. During days 300-350, for example, large fluctuations in temperature occurred at

the mid-bay station (Fig. 6b), relative to the bay entrance (Fig. 6a) or the upper bay (Fig. 6c). These spatio-

temporal variations were simulated well by the model, demonstrating not only the good performance of the

model but also the reliability of the atmospheric forcing data. The model performance in reproducing

temperature over the Texas-Louisiana shelf was further examined using the satellite data for sea surface

temperature (SST). Seasonality of the SST extracted from MODIS over the northwestern GoM is overall

reproduced well (Fig. 7). It is worth noting that the model also reproduces the relatively low temperatures on the

southern Texas coast during summer, which is a well-known upwelling zone during the summertime when

upcoast (from Texas toward Louisiana) winds force an offshore surface transport (Zavala-Hidalgo et al., 2003).



### 3.4 Shelf current

One of the important features of the Texas-Louisiana shelf is the quasi-annual pattern of shelf current, which is predominantly westward most of the time except during summer (Cochrane and Kelly, 1986; Li et al., 1997; Cho et al., 1998). The prominent downcoast (from Louisiana toward Texas) shelf current is driven by along-shelf wind and enhanced by the Mississippi-Atchafalaya river discharge (Oey, 1995; Li et el., 1997; Nowlin et al., 2005). Under summer wind that usually has an upcoast component, the nearshore flow is reversed to upcoast direction (Li et al., 1997). The model reproduces well the observed subtidal component of surface along-shelf current at two TABS buoy stations outside of Galveston Bay, buoy B (~20 km offshore) and buoy F (~80 km offshore) (Fig. 8). The overall reproduction of the subtidal shelf current is satisfactory, with the *MAE* of 8-14 cm s$^{-1}$ and the model skill of 0.67-0.88 (Table 1). Note that both the observed and modeled along-shore shelf currents are mostly westward, except during summer, consistent with the previous observations (e.g., Cochrane and Kelly, 1986).

### 4. Remote influence from shelf current and major rivers

Texas-Louisiana coast is characterized by broad and relatively shallow shelf receiving river discharges from multiple rivers (Fig. 1). The river plumes from nearby bays are likely to interact with each other on the shelf. For example, superimposition of multiple river discharges on the shelf will lead to a plume with a larger scale (Warrick and Farnworth, 2017). River discharges from major rivers into the Texas-Louisiana shelf (Fig. 1) vary in their magnitudes from 10 to 10$^4$ m$^3$ s$^{-1}$, which may result in plumes in a variety of temporal and spatial scales. Plumes from larger rivers tend to persist for a longer time and extend over a larger area. In addition to the river discharge, the extent of river plume is also subject to modulation by shelf current. Shelf current on the Texas-Louisiana shelf has been observed to vary seasonally depending on the along-shore wind as well as the Loop Current eddies (DiMarco et al., 2000; Ohlmann and Niiler, 2005). River plumes are closely related to the distribution of freshwater and riverine materials including organic and inorganic nutrients, sediments, and pollutants on the shelf (Turner et al., 2007; Brodie et al., 2012; Horner-Devine et al., 2015). Horizontally, their distribution influences but is also regulated by the shelf dynamics, and exhibits significant seasonal variation. Vertically, the distribution of fresher water over the shelf is the result of the competition between freshwater buoyancy and vertical mixing (Fong and Geyer, 2001).

The conditions in Texas coastal waters, however, are impacted by several remote sources, including mesoscale eddies (Oey et al., 2005; Ohlmann and Niiler, 2005), along-shore transport of low-salinity water from major rivers (Li et al., 1997; Nowlin et al., 2005), and Ekman transport induced by longshore wind and the



resulting upwelling/downwelling (Li et al., 1997; Zhang et al., 2012). Here, we discuss the remote influences of

major river plumes and shelf dynamics on the salinity regime along the Texas coast.

**4.1 Influence of shelf current**

The strength and direction of shelf current are sensitive to the wind field. Generally, residual current

over the Texas-Louisiana shelf has a significant downcoast component during non-summer seasons and an

upcoast component during summer season, mainly because of the seasonal variation of wind field (Oey et al.,

2005). The seasonally varying shelf current affects the distribution of lower-salinity water. Comparison of the

model results on day 150 (May 31, 2007) and day 160 (June 10, 2007) clearly shows the different distribution

of lower-salinity water along the coast in response to wind field and the resulting shelf current (Fig. 9). The

river discharge differences between these two days are negligible and thus the differences in lower-salinity

water distribution can be mainly attributed to the differences in shelf current. Day 150 was characterized by a

significant downcoast shelf current in the inner shelf, with a current speed exceeding 0.5 m s$^{-1}$, while day 160

was characterized by a rather weak shelf current with a speed less than 0.1 m s$^{-1}$. The pattern of the surface

residual current is related to the wind field. At day 150, a downcoast component of the wind induced an onshore

Ekman transport, which in turn resulted in a downcoast geostrophic flow (Li et al., 1997). This downcoast flow

transported fresher Mississippi-Atchafalaya water toward Texas while constraining it on the inner shelf and thus

forming a narrow band of lower-salinity water close to the shoreline (Fig. 9e). Under weak or upcoast shelf

current, on the other hand, this constraining was weakened, forming a wider band of lower-salinity water (Fig.

9f). Also note that salinity on the Texas inner shelf was higher on day 160 than that on day 150.

Regulated by the shelf current, salinity distribution over the shelf exhibits evident seasonality. The

modeled salinity distribution shows that the lower-salinity water extended toward the western Texas inner shelf

until May (Fig. 10). This was in response to the variation in the Mississippi-Atchafalaya discharge that

increased from January and peaked in late April in 2008 (Mississippi discharge data at

https://waterdata.usgs.gov/usa/nwis/uv?site_no=07374000). The salinity at the Galveston Bay mouth decreased

by about 10 psu from January to May. Then, starting from June, the salinity along western Texas shelf gradually

increased as the higher-salinity water moved upcoast, which was evident in both the observation and model

results during summer (Fig. 8). The salinity at the Galveston Bay mouth increased from less than 20 psu in June

to >30 psu in August (Fig. 10), about the same magnitude of salinity change from January to May. It suggests

that the influence on the salinity in the Texas coast from the seasonally varying shelf circulation may be

comparable to that from the seasonal variation in the Mississippi-Atchafalaya discharge.





## 4.2 Remote influence from Mississippi-Atchafalaya river discharge


The results from three numerical experiments show that the daily mean surface salinity time-series at the mouths of Galveston Bay and Aransas Bay show different time scales with which the salinity responds to the Mississippi-Atchafalaya discharge (Fig. 11). At the Galveston Bay mouth, the salinity begins to decrease from about day 25 in response to the Mississippi-Atchafalaya discharge and continues to decrease until around day

150 when it reaches a quasi-steady state. It takes about 65 days for the Mississippi-Atchafalaya discharge to start to decrease the salinity at the Aransas Bay mouth, which continues until day 190. The long-term mean discharge from the Mississippi and Atchafalaya rivers reduces the salinity by about 10 psu (the difference between the cases Jan-G and Jan-GAM) at the Galveston Bay mouth and 8 psu at the Aransas Bay mouth under the typical winter wind condition.

What is interesting at the Galveston Bay mouth is that the magnitude of the salinity drop varies greatly between the January and July winds, with the salinity decrease under the January wind about 5 psu larger compared to that under the July wind. Further south at the Aransas Bay mouth, salinity changes little in response to the discharges from Galveston Bay and the Mississippi-Atchafalaya rivers under the July upcoast wind condition. The influence from Galveston Bay is very limited at the Aransas Bay mouth even under a

downcoast wind; it is reasonable to assume the influence will be even smaller under an upcoast wind. Surface salinity maps averaged over days 250-300 show distinctly different spatial extent of the lower-salinity water originated from the Mississippi-Atchafalaya rivers and Galveston Bay under different wind conditions (Fig. 12). Under the January wind, the lower-salinity water is trapped nearshore by the shelf current, forming a narrow band along the coast, similar to the distribution on day 150 of the two-year realistic model run (Fig. 9e). On the

other hand, under the July wind, water on the Texas shelf is efficiently replenished by the saltier water originated from the southwest, forming a tongue-shaped intrusion of the saltier water toward the Louisiana coast and a much wider lower-salinity band on the Louisiana shelf (e.g., using 30 psu as a criterion). Consequently, salinity is higher on the Texas shelf and lower on the Louisiana shelf when compared to that under the January wind. This higher salinity on the Texas shelf during summer is consistent with the realistic model run (Fig. 10).

Salinity change due to the remote river input and shift in wind field affects the estuarine dynamics, such as estuarine circulation, salt flux, and estuarine-coastal water exchange. We examined the difference in salt flux and exchange flow at Galveston Bay due to remote river influence and different shelf current. Following Lerzak et al. (2006), we calculated the tidally averaged and cross-sectionally varying components ($u_e$ and $S_e$) from the normal velocity $u$ and salinity $S$. From the vertical profiles of $u_e$ and $S_e$ at the deepest part of the bay mouth, it is

evident that in the lower layer $u_e$ is strongest (maximum of 6 cm s$^{-1}$) and $S_e$ is largest (maximum of 0.95 psu) for the case Jan-G, indicating the strongest exchange flow (i.e., estuarine circulation), compared to the other two





cases with the Mississippi-Atchafalaya discharge (Fig. 13). On the contrary, the case Jan-GAM shows the weakest bottom inflow (maximum of 4 cm s$^{-1}$) and the smallest bottom $S_e$ (maximum of 0.60 psu). The Mississippi-Atchafalaya discharge under the January wind condition decreases the salinity at the bay mouth,

thus resulting in a weaker horizontal salinity gradient and a weaker exchange flow.

The influence of the Mississippi-Atchafalaya discharge on the dynamics of Galveston Bay was further examined using total exchange flow (TEF) that quantifies the overall change in estuarine water renewing efficiency. Using the isohaline framework method proposed by MacCready (2011), which was found to be a precise way to calculate the landward transport (Chen et al., 2012), we estimated the total exchange flow ($Q_{in}$,

the flux of water into the estuary due to all tidal and subtidal processes), the resulting salt flux into the estuary ($F_{in}$), the mean residence time ($T_{res}$, the ratio of salt mass inside the estuary to $F_{in}$). Table 2 lists the values of $Q_{in}$, $F_{in}$, and $T_{res}$ for the three idealized model runs. For the exchange flow, $Q_{in}$ is largest for the case Jan-G and smallest for the case Jan-GAM. The Mississippi-Atchafalaya discharge under the January wind condition lowers the salinity at the Galveston Bay mouth by 10 psu (Fig. 11a), effectively slowing down the water

exchange between the bay and coastal ocean. The reduction in $Q_{in}$ caused by the remote discharge (470 m$^3$ s$^{-1}$ = 24% reduction) is larger than the long-term mean river input into Galveston Bay (350 m$^3$ s$^{-1}$; Du et al., 2019). Moreover, $F_{in}$ for the case Jan-GAM is about half of that under the Jan-G. As a result, $T_{res}$ of the bay is largest in the case Jan-GAM, although the difference in $T_{res}$ is not as large as that in $F_{in}$ because the bay has the smallest salt mass in the case Jan-GAM (Table 2). This analysis also suggests that the exchange between the

bay and coastal ocean is likely stronger during summer than that during winter under the same river discharge condition.

## 5. Summary

An unstructured-grid hydrodynamic model with hybrid vertical grids was developed for Galveston Bay and the shelf and validated for water level, current velocity, salinity, and temperature for Galveston Bay as well

as over the shelf. The good model performance, particularly in terms of salinity (vertically/horizontally), is at least in part attributable to the inclusion of multiple river plumes along the shore as well as the interaction between estuaries and shelf. This model provides a good platform that can be used for other purposes in future studies. Its flexibility in the horizontal and vertical grids allows refinement in any region of interest without penalty in the time step (due to the semi-implicit scheme). For example, it would be relatively easy to adapt the

model by refining the grid inside any target bay, e.g., Corpus Christi Bay.

We demonstrate the necessity of including the Mississippi and Atchafalaya rivers for the modeling of the Texas coastal systems, particularly for the processes associated with relatively long time scales (e.g.,

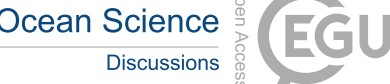



months). Receiving relatively small freshwater discharge and limited by narrow outlets and small tidal ranges, the estuarine bay systems along the Texas coast, e.g., Galveston Bay, Aransas Bay, and Corpse Christi Bay, are

characterized by relatively slow water exchange and long flushing times. In this paper, we show that the exchange flow plays an important role for the water renewal and that the exchange flow varies greatly depending on the wind field and the resulting shelf currents. Modulation in discharge of the Mississippi-Atchafalaya rivers, when coupled with downcoast wind conditions, could have a great influence on the dynamics of estuaries along the Texas coast.

**Author contributions**

J. Du and K. Park lead the effort of model development, data analysis, and preparation of the manuscript. J. Shen, Y. J. Zhang, F. Ye, and Z. Wang provided guideline for the model configurations in terms of preparation of forcing files and boundary conditions. X. Yu assisted the visualization of modeled and observed data. N. N. Rabalais provided the shelf-wide survey data for the model validation. All authors were

involved in the manuscript writing.

**Acknowledgements:**

All the observational data used for model validation are available online. Salinity data are extracted from TDWB (https://waterdatafortexas.org/coastal). Contiguous monitoring data of temperature and water level are extracted from NOAA Tide and Current (https://tidesandcurrents.noaa.gov/). Surface buoy current data are

extracted from TABS (http://pong.tamu.edu/tabswebsite/). Daily satellite data (4 km resolution) are extracted from (https://podaac.jpl.nasa.gov/). Shelf-wide summer survey data of 2008 is accessible at NODC with the accession number of 0069471 (https://www.data.gov/). The numerical simulation was performed on the high-performance computer cluster at the College of William and Mary.

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



**Table 1**: Error estimates for model-data comparison for 2007-2008.

| Variables | Station | Total | | Subtidal | |
|---|---|---|---|---|---|
| | | MAE | Skill | MAE | Skill |
| **Water level (cm)** | Morgan's Point | 9 | 0.92 | 6 | 0.96 |
| | Eagle's Point | 7 | 0.96 | 6 | 0.96 |
| | Bay Entrance | 9 | 0.95 | 6 | 0.95 |
| | Freeport | 8 | 0.96 | 6 | 0.95 |
| | Bob Hall | 7 | 0.96 | 5 | 0.94 |
| | Pilot Station | 11 | 0.86 | 6 | 0.85 |
| | Dauphin Island | 7 | 0.94 | 7 | 0.87 |
| **Salinity (psu)** | TRIN (1.5 m)[a] | 2 | 0.93 | 2 | 0.93 |
| | BAYT (2.0 m)[a] | 2 | 0.89 | 2 | 0.89 |
| | MIDG (3.1 m)[a] | 4 | 0.87 | 2 | 0.87 |
| | BOLI (2.9 m)[a] | 2 | 0.76 | 3 | 0.69 |
| **Temperature (°C)[b]** | Morgan's Point | 1 | 0.99 | 1 | 0.99 |
| | Eagle's Point | 1 | 0.99 | 1 | 0.99 |
| | Bay Entrance | 1 | 0.97 | 1 | 0.98 |
| **Velocity (cm sec$^{-1}$)[b]** | Buoy B | 14 | 0.88 | 11 | 0.82 |
| | Buoy F | 10 | 0.79 | 8 | 0.67 |

[a] The value within the parenthesis indicates the mean depth below surface of the sensor.
[b] Temperature and velocity data are surface data.


**Table 2**: Total exchange flow ($Q_{in}$) and the resulting salt flux ($F_{in}$) at the Galveston Bay mouth, and the mean residence time of the bay ($T_{res}$) based on the isohaline method in MacCready (2011).

| Case ID[a] | $Q_{in}$ (m$^3$ s$^{-1}$) | $F_{in}$ (kg salt s$^{-1}$) | $T_{res}$ (days) | $S_{mean}$[b] (psu) |
|---|---|---|---|---|
| Jan-G | $1.93\times10^3$ | $6.75\times10^4$ | 13.0 | 20 |
| Jan-GAM | $1.46\times10^3$ | $3.47\times10^4$ | 16.0 | 13 |
| Jul-GAM | $1.80\times10^3$ | $5.30\times10^4$ | 13.1 | 16 |

[a] see Fig. 12 for the explanation of idealized runs
[b] Mean salinity (volume-weighted average over days 250-300) inside the bay



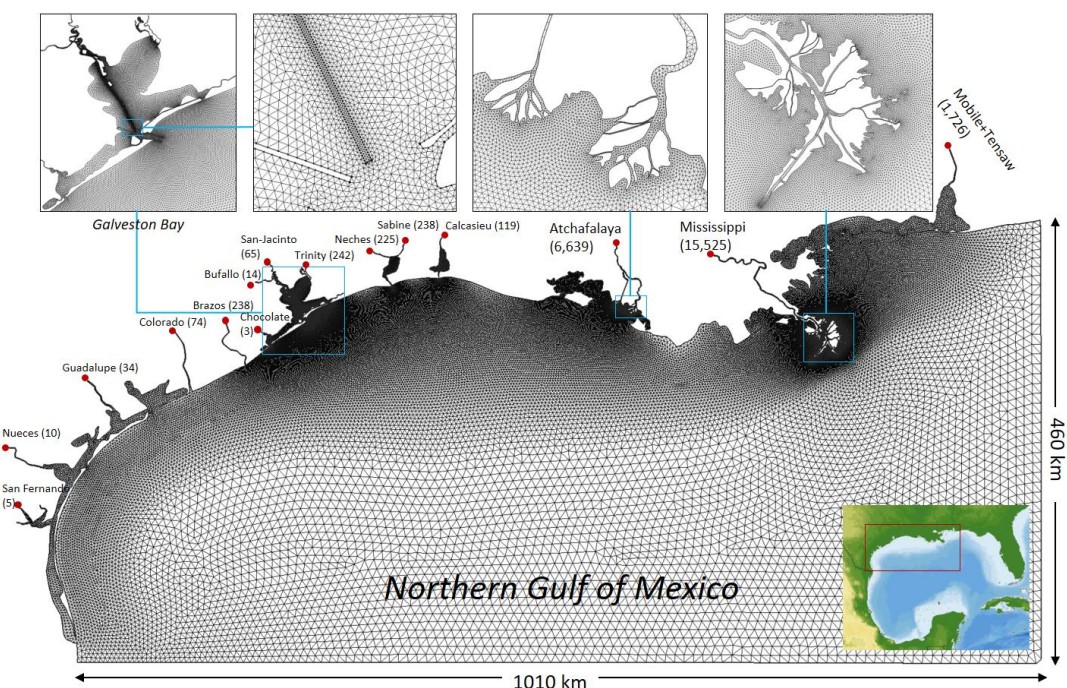

**Figure 1**: The model domain and the horizontal grid, with the upper panels showing zoom-ins of selected coastal systems. Locations of major river inputs are indicated with red dots, with the associated mean river discharges ($m^3 \ s^{-1}$) shown in the parentheses.





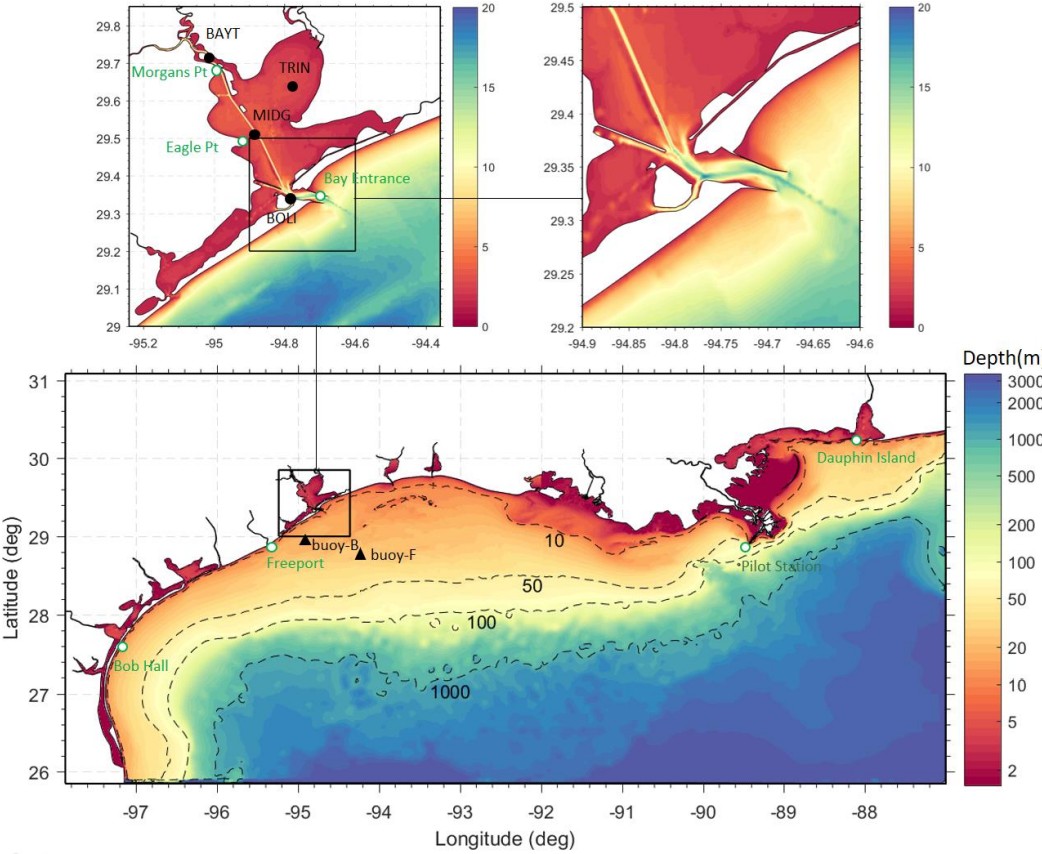

**Figure 2**: Bathymetry in the model domain with the upper panels showing zoom-ins of Galveston Bay and its main entrance. Note that the lower panel uses the log scale for depth because of a very wide range of depth over the entire model domain. Also shown are the NOAA tidal gauge stations (open green circles), TWDB (Texas Water Development Board) salinity monitoring stations (solid black circles), and TABS (Texas Automated Buoy System) buoy stations (black solid triangles).




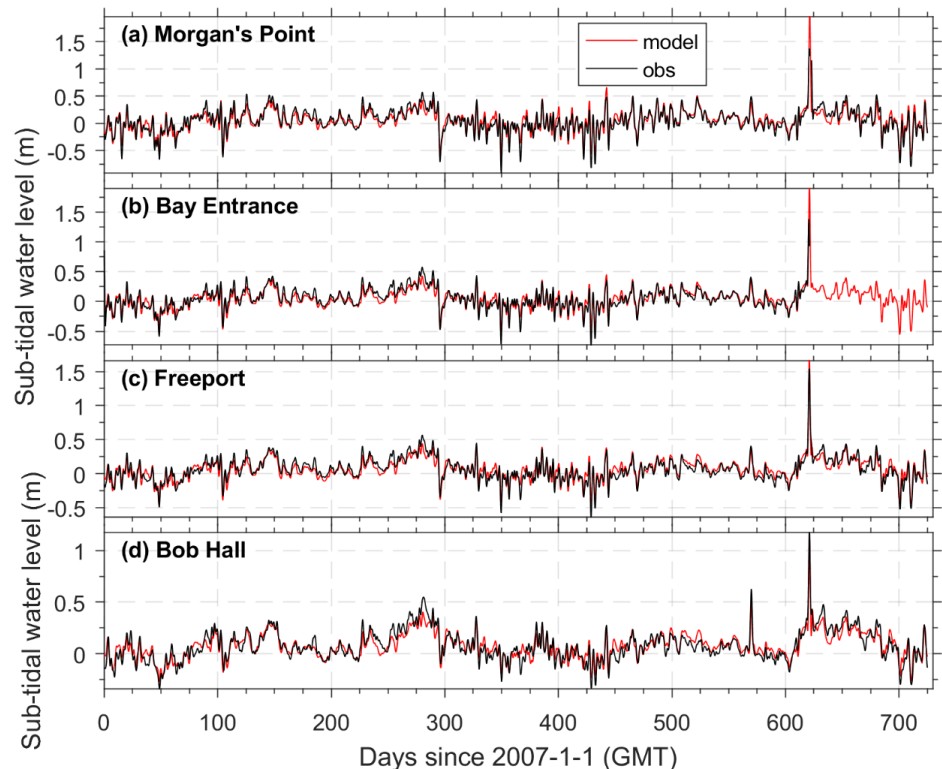

**Figure 3**: Subtidal surface elevation comparison between model (red line) and observation (black line) at NOAA tidal gauge stations (see Fig.2 for their locations).





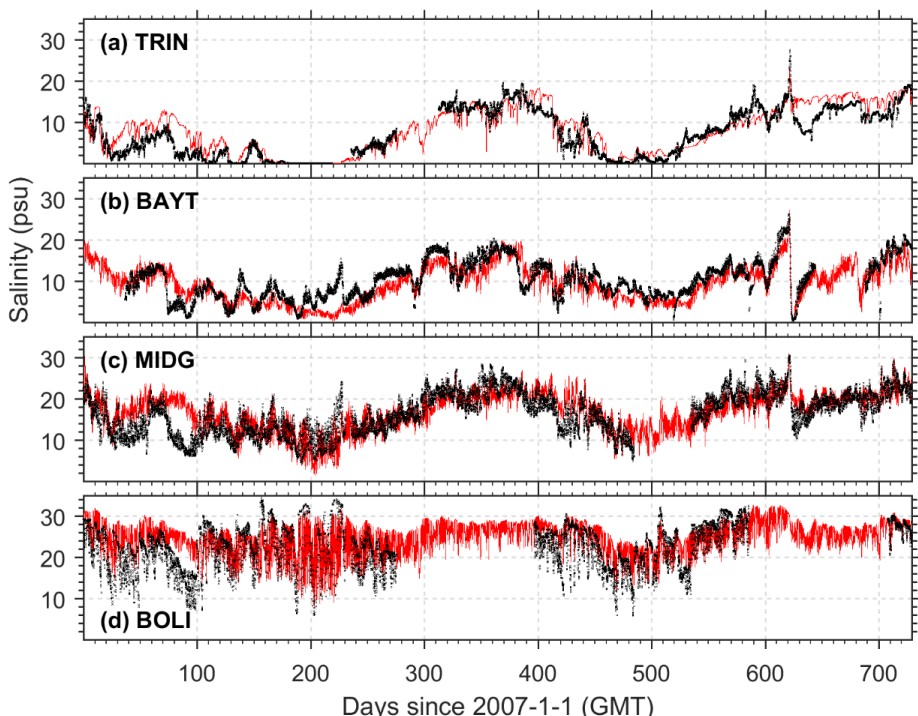

**Figure 4**: Salinity comparison between model (red line) and observation (black cross) at four TWDB stations (see Fig. 2 for their locations).






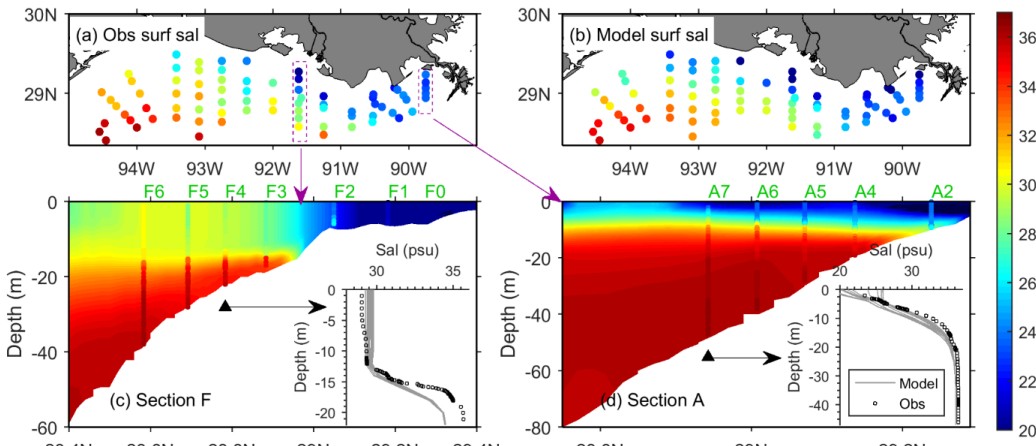

**Figure 5**: Salinity distribution at the Louisiana-Texas shelf from the shelf-wide survey on July 22-27, 2018: comparison of (a) observed and (b) modeled surface salinity and of the vertical profiles at two cross-shelf sections (c) F and (d) A. In (c) and (d), the colored dots indicate observed salinity while the filled colors indicate modeled salinity, and the insets compare the vertical profiles of salinity at the selected stations of F4

and A7, respectively. The model in the insets show 12 modeled profiles over one day (the observation time ± 0.5 day).



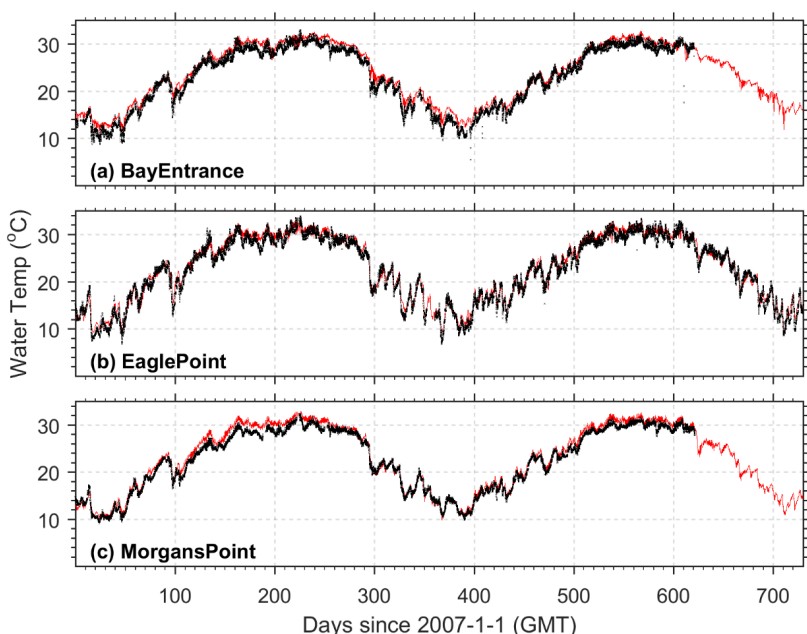

**Figure 6**: Temperature comparison between model (red line) and observation (black line) at three NOAA
stations (see Fig. 2 for their locations).




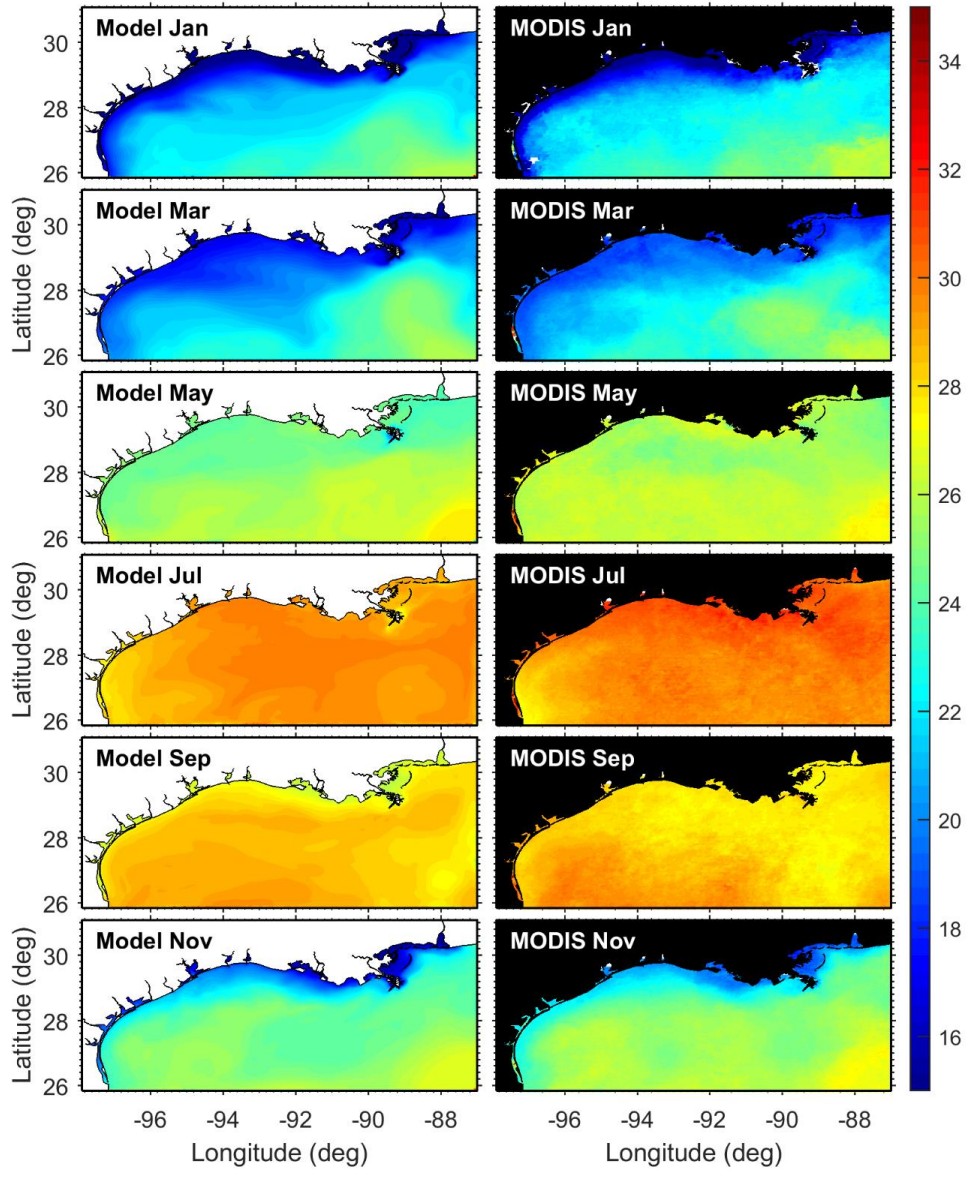

**Figure 7**: Temperature comparison (monthly average) between model (left panels) and MODIS satellite data (right panels) for selected months in 2008.






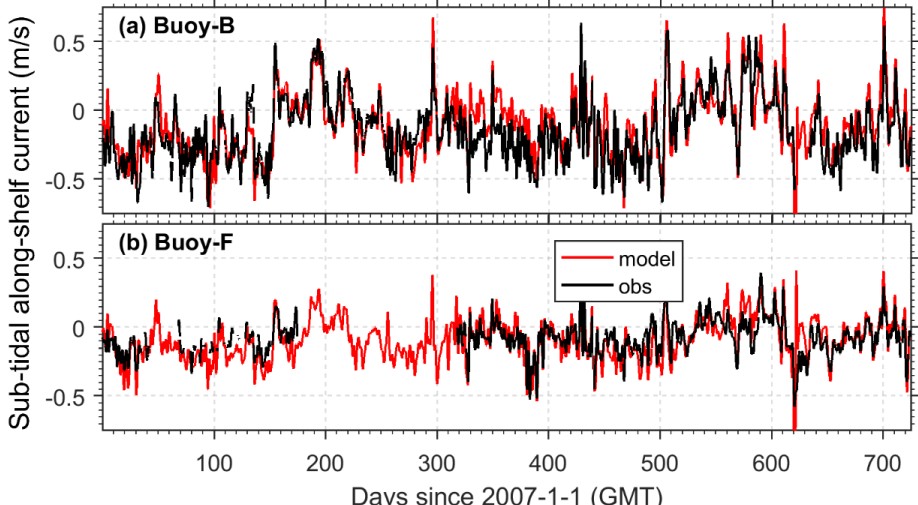

**Figure 8**: Comparison of the subtidal east-west surface shelf current between model (red line) and observation (black line) at two TABS buoys (see Fig. 2 for their locations).




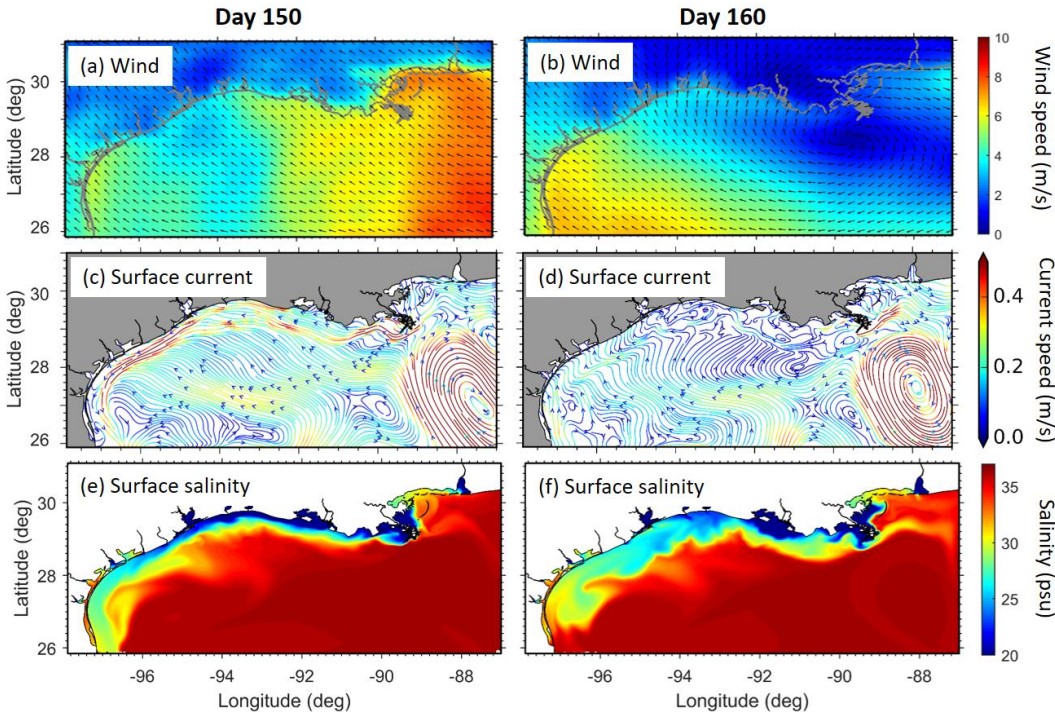

**Figure 9**: Comparison of the observed wind field and the modeled surface residual current and surface salinity
on day 150 (May 31, 2007) and day 160 (June 10, 2017). The filled colors indicate the daily mean wind speed
(a-b), speed of residual current (c-d) and salinity (e-f).



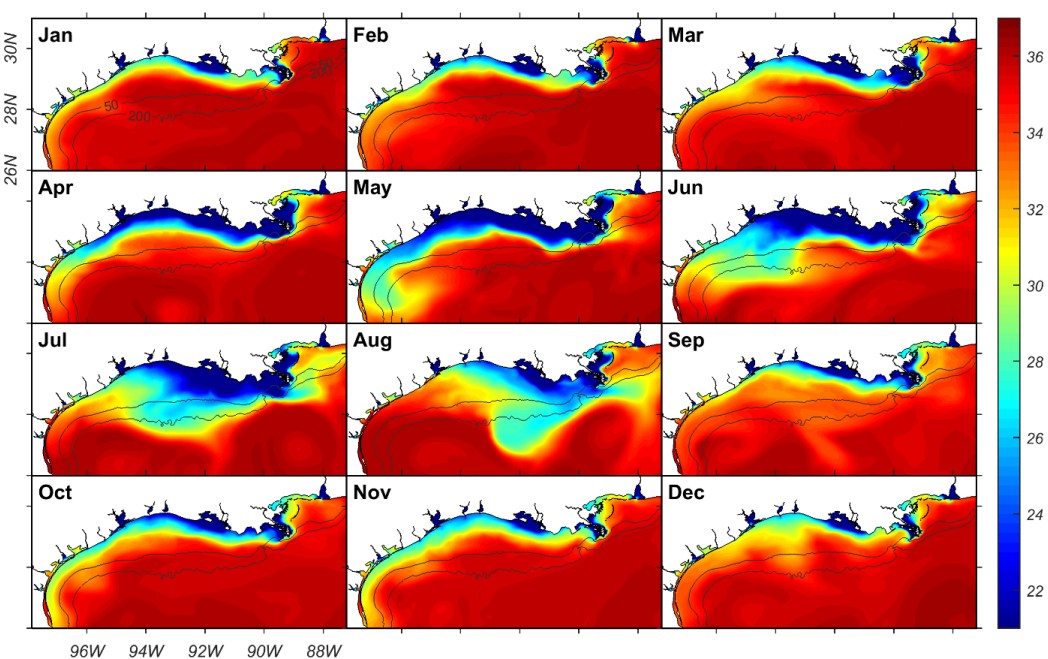

**Figure 10**: The modeled monthly mean surface salinity in 2008, with the black contour lines denoting the depth contours of 50 and 200 m.





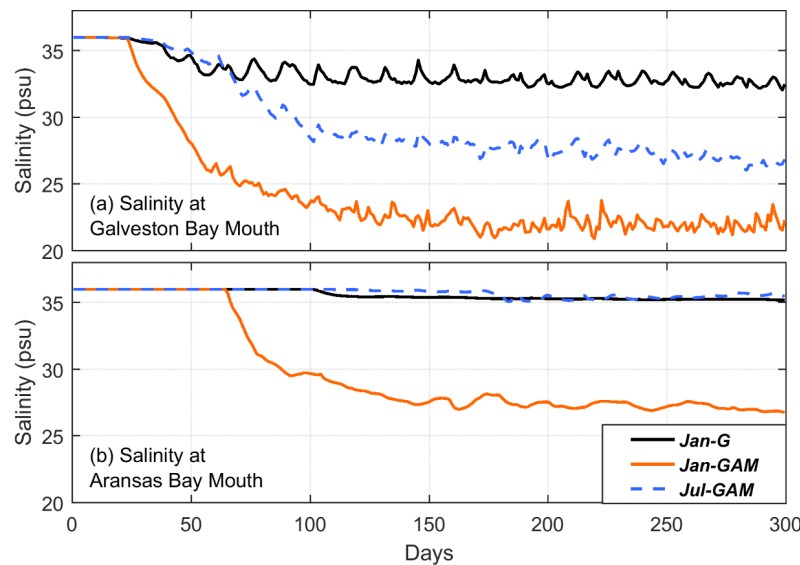

**Figure 11**: Daily mean surface salinity at the mouth of (a) Galveston Bay and (b) Aransas Bay for three idealized model runs with constant long-term mean river discharges: river discharges into Galveston Bay only with January 2018 wind field (Jan-G) and the Mississippi-Atchafalaya river discharge as well as discharges into Galveston Bay with January 2018 wind field (Jan-GAM) or July 2018 wind field (Jul-GAM). The January or July wind field was repeated every month to take into account the wind variability. Starting with the spatially uniform constant salinity of 36 psu throughout the model domain, each model run shows the effect on salinity dilution under the specified condition of river discharge and wind.


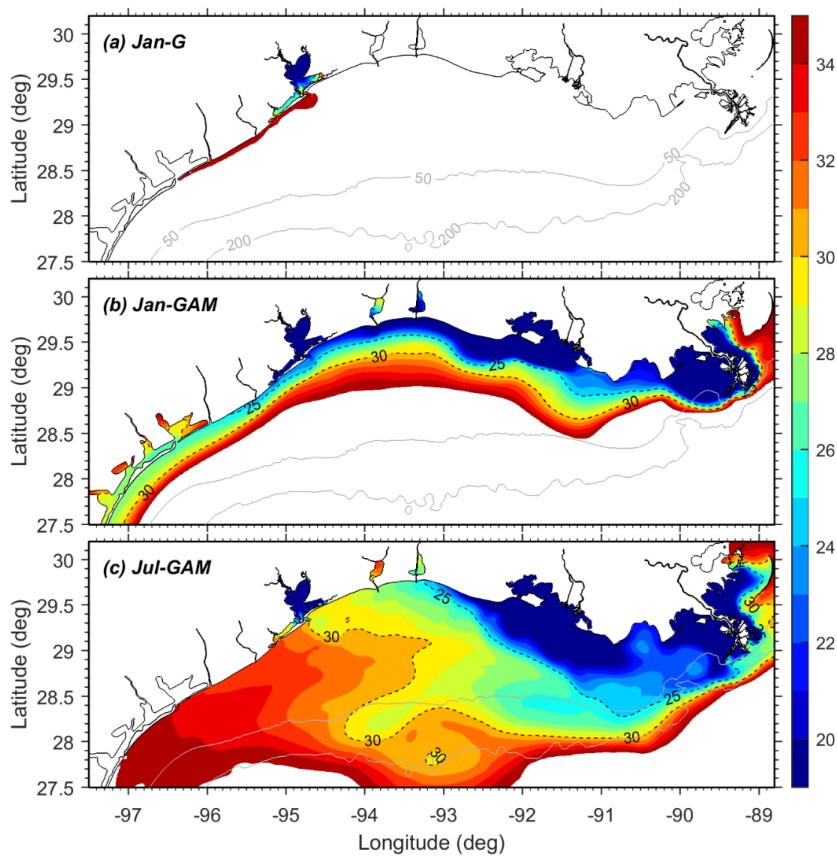

**Figure 12**: Surface salinity distributions, averaged over days 250-300, from three idealized model runs (see Fig. 11 for the explanation of idealized runs). Values larger than 35 psu are shown as blank and the grey contour lines denote the depth contours of 50 and 200 m.






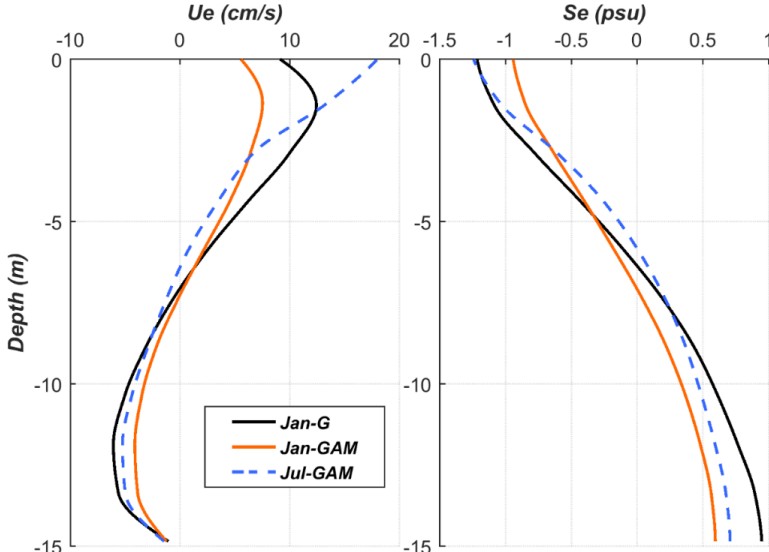

**Figure 13**: Vertical profiles of exchange flow ($u_e$) and salinity ($S_e$) at the deepest part of the Galveston Bay mouth, averaged over days 250-300, for three idealized runs (see Fig. 11 for the explanation of idealized runs). The positive (negative) $u_e$ is out of (into) the bay.
