# Peer review of "A hydrodynamic model for Galveston Bay and the shelf in the northern Gulf of Mexico"

_Ocean Science, 2019_

## Referee Comment (RC1) · Anonymous Referee #1 · 14 Mar 2019

General Comments:

The manuscript " A hydrodynamic model for Galveston Bay and the shelf in the northwestern Gulf of Mexico" by Du et al. presents a detailed study on development, validation and implementation of an unstructured grid model in the Gulf of Mexico and adjacent estuarine bays. The impact of the remote river discharge from Mississippi and Atchafalaya on the hydrodynamics in the Galveston Bay is assessed. Overall, the model was well established, and carefully validated. The model results are quite reliable. A large-scale unstructured model is very helpful in identifying the effect of shelf processes on a specific estuary. The unstructured model provides great flexibility in covering the large GOM domain and resolving the complex shoreline and bathymetry in the estuaries. This study presents a good example of research on estuary-shelf interaction. The manuscript is well written and organized, and several new findings are provided, thus it merits a publication after some minor revision. My main suggestions are: 1) More focus could be paid on the physics, like how the buoyancy induced coastal current from the Atchafalaya and Mississippi Rivers affects the advective transport and mixing at the mouth of the Galveston Bay; 2) How about the advantage of this unstructured grid in resolving the narrow and deep channel? How important is this resolution in reproducing salt intrusion? Based on my previous experience, this model seems a little difficult to simulate salt intrusion in the Pearl River Estuary. The model results always seem overmixed than observations; 3) How Galveston and other estuaries affect the shelf processes, e.g., how they impact the coastal current, and water column stratification and mixing?

Detailed comments:

No

---

## Referee Comment (RC2) · Anonymous Referee #2 · 18 Mar 2019

Manuscript entitled "A hydrodynamic model for Galveston Bay and the shelf in the northwestern Gulf of Mexico" by Jiabi Du et al. presents study of the influences and effects that seasonal wind forcing has on the salinity distribution along the Louisiana shelf with focus on Galveston Bay. As this problem is having multi-scale dependency i.e. shelf dynamics has important influence on the coastal dynamics and vice versa, unstructured modelling approach seems to give reasonable answers and is appropriate. Manuscript is well organised, with some additional information should be valid contribution and appropriate for the journal.

General comments: Study is covering big portion of work done, however I think readers would benefit from clear and possibly additional explanation of baseline method used in the study. In other words, it is not clear to me if authors assumed and explored

exclusively wind (January vs. July) effects in the 3 numerical simulations (Jan-G, Jan-GAM, July-GAM) using the same wind field (Jan in first 2 or July in 3rd) replicated in time during the whole year without using any heat flux (or other atmospheric model forcing) or tidal forcing. One table listing used/or not used assumptions would help (i.e. no heat flux, no boundary conditions from Hycom, no tides, initial field, winds from Jan or July replicated during the whole simulation). If this is the case (when there is no heat flux, but initial stratification), then the simulation represent only partly barotropic approach which is valid in shallow part of the domain and during the winter only - as there is no vertical heat flux supporting vertical stratification in balance with vertical mixing parametrised with turbulence. I would be surprised that model didn't vertically mixed the whole vertical column as 1 year of simulation is quite enough time. In the case they used vertically uniform density field for start then I have doubts it does represent valid approximation of GOM in July. Possibly, then more correct title of the experiments would be to call (and explain) those sensitivity experiments as sensitivity of the salinity field to the wind field effects (and not mix that with July/January as seasons) of the simple barotropic system interacting with coastline and bathymetry. In that case dynamics will be only due to buoyancy effects of the rivers via salinity and some wind mixing/transport without any temperature variations. Validation period using full forcing is then confirmation of model setup and tunning. Part with residence time (336- 351) seems as added to the manuscript without needed explanation of method MacCready (2011). Authors could at lest give basic equations for completeness of the study and to show how they computed values.

Specific comments: In Abstract; 1) I think the main mesage is to present RESULTS of the study using 3D SCHISM and not to present model (first sentence)? 2) If they used only Hycom boundary conditions then it is global model and not models (line 16), or if they used added tidal elevations then should state that precisely.

in Methodology (2.1): 1) line 90: Does Schism use simple 1 order Galerkin method for momentum of higher order (as it does for tracers)? If not does the authors think this is

not relevant for the study where wind dynamics and momentum plays important role?

in Forcing conditions (2.3): 1) line 124: model or models? 2) line 142: what was used to compute heat/momentum/fresh water processes between ocean and atmosphere? If this is bulk flux then they should reference. 3) line 146: definition of sub-tidal period for boundary condition filtering was set to 15 days and later in the text they use 2 days? Is there particular reason why they chose 15 days and not less (i.e. 2 days) which would allow for inclusion of eddy dynamics embedded in Hycom model?

in Numerical experiments (2.4): 1) line 150-152: Authors used constant and the same river flux in Galveston Bay during the whole year in all 3 experiments? Did they used the same and constant fluxes for GAM in experiments Jan-GAM and July-GAM? What were the values? 2) line 156: methodology of replicating January wind during the whole year is a bit strange; as it captures some variability within month that is replicated 12 times. What would make more sense is to use "typical winter / summer" case where they could compute multi-year mean wind field from ECMWF fileds. Otherwise January/July as generic names have different meaning (authors used specific 2008 winds so they are not really generic i.e. seasonal in strict definition). 3) line 160: I am not sure what authors mean with computing boundary conditions form 2 years temporally constant?

in Water level (3.1): 1) in line 180: Why not to state what is the Cd equivalent to Manning coef as authors used quadratic bottom friction, instead of reporting Manning's coef? What was the method and how they tuned Manning coef is not really clear. 1) line 185: They speculate that low skill at Pilot Station is due to proximity of boundary conditions, which seems not plausibly as boundary conditions are fare away. Another point is that low-frequency MAE (i.e. boundary effect contribution) is much better than total which implies that other dynamics is important contribution to the MAE (Table 1)?

in Salinity (3.2): line 204: How authors explain lower MAE for global than low-passed filtered case in BOLI station? This seems hard to believe in mathematical sense, possibly some error.

in 4. (Remote influence): line 263-264: Sentence is not clear and make no sense: "Horizontally, their distribution influences but is also regulated by the shelf dynamics, and exhibits significant seasonal variation." lines 336-351 should include equation how they computed residence time.

in 5. (Summary): I think that authors should emphasise main results from their study and answers they provided on questions posed in the last paragraph of introduction (i.e. time needed for information originating at Mississippi-Atch rivers to arrive to GB? About extent and portion of seasonal influence of winds to the horizontal distribution of salinity etc.). This way written summary seems too short and doesn't summarise the study.

---

## Author Comment (AC2) · 17 Apr 2019

**Response to comments by reviewer #2**

Manuscript entitled "A hydrodynamic model for Galveston Bay and the shelf in the northwestern Gulf of Mexico" by Jiabi Du et al. presents a study of the influences and effects that seasonal wind forcing has on the salinity distribution along the Louisiana shelf with focus on Galveston Bay. As this problem is having multi-scale dependency, i.e. shelf dynamics has important influence on the coastal dynamics and vice versa, unstructured modelling approach seems to give reasonable answers and is appropriate. Manuscript is well organized, with some additional information should be valid contribution and appropriate for the journal.

Thanks very much for your comments, which have helped clarify multiple potentially confusing statements in the manuscript.

General comments:

1) Study is covering big portion of work done, however I think readers would benefit from clear and possibly additional explanation of baseline method used in the study. In other words, it is not clear to me if authors assumed and explored exclusively wind (January vs. July) effects in the 3 numerical simulations (Jan-G, Jan-GAM, July-GAM) using the same wind field (Jan in first 2 or July in 3rd) replicated in time during the whole year without using any heat flux (or other atmospheric model forcing) or tidal forcing. One table listing used/or not used assumptions would help (i.e. no heat flux, no boundary conditions from Hycom, no tides, initial field, winds from Jan or July replicated during the whole simulation).

The numerical experiments have virtually the same setting as in the realistic 2007-2008 model run, including tide, heat exchange, and ocean boundary conditions. The only differences from the realistic run are: (1) an initial salinity condition of 36 psu over the entire domain; (2) using repeated monthly wind forcing of January or July 2008; (3) long-term mean constant river discharges from three estuarine systems (Mississippi, Atchafalaya, and Galveston Bay). The table below shows the differences in numerical experiments relative to the realistic run. With only three differences, we do not plan to add this table in the revised manuscript. Instead, we will add additional sentences describing the setting of the numerical experiments in the revised manuscript.

Table A: Settings for the numerical experiments.

|  | Jan-G | Jan-GAM | Jul-GAM |
|---|---|---|---|
| River discharge (m3/s) | Q(Trinity River)=190 Q(San Jacinto)=52 Q(Buffalo Bayou)=14 Q(Chocolate Bayou)=3 | Q in Jan-G, plus Q(Atchafalaya River)=15525 Q (Mississippi River)=6664 | |
| Wind | Repeated wind of January 2008 | | Repeated wind of July 2008 |
| Initial condition for salinity | 36 psu over the entire domain throughout the water column | | |
| Initial condition for temperature | Same as in realistic model run, based on HYCOM output on 2007-01-01 | | |

| Air temperature, pressure, solar radiation | Same as in realistic model run, spatially and temporally varying from 2007-01-01 |
|---|---|
| Ocean boundary condition | Same as in realistic model run, spatially and temporally varying from 2007-01-01 |

2) If this is the case (when there is no heat flux, but initial stratification), then the simulation represent only partly barotropic approach which is valid in shallow part of the domain and during the winter only - as there is no vertical heat flux supporting vertical stratification in balance with vertical mixing parametrized with turbulence. I would be surprised that model didn't vertically mixed the whole vertical column as 1 year of simulation is quite enough time. In the case they used vertically uniform density field for start then I have doubts it does represent valid approximation of GOM in July. Possibly, then more correct title of the experiments would be to call (and explain) those sensitivity experiments as sensitivity of the salinity field to the wind field effects (and not mix that with July/January as seasons) of the simple barotropic system interacting with coastline and bathymetry. In that case dynamics will be only due to buoyancy effects of the rivers via salinity and some wind mixing/transport without any temperature variations. Validation period using full forcing is then confirmation of model setup and tunning.

The heat flux is included in numerical experiments. We will clarify this in the revised manuscript. The temperature profile at a station in the deep Gulf (Fig. A) indicates that in this baroclinic model run, the deep ocean is persistently stratified, with the surface temperature changing seasonally. The stratification at the bay mouth also persists (Fig. B).

[Figure]

Figure A: The temperature profile (low-pass filtered with a cut-off period of 50 h) at a station in the deep Gulf (see the inset for the station location): the grey lines in the inset denoting the 50, 100, 150, and 200 m bathymetric contours. (Note: we do not plan to include this figure in the revised manuscript).

[Figure]

Figure B: (a) Surface salinity at the mouth of Galveston Bay (the bold lines indicating the sub-tidal salinity) and (b-d) the vertical salinity profiles at the mouth (low-passed filtered with a cut-off period of 50 h). The periodicity in (b-d) is the tropic-equatorial cycle. Note: we do not plan to include this figure (showing the water column is not well mixed in this a full 3D baroclinic mode run) in the revised manuscript.

3) Part with residence time (336-351) seems as added to the manuscript without needed explanation of method MacCready (2011). Authors could at least give basic equations for completeness of the study and to show how they computed values.

The equations for the salt flux including the salt flux decomposition and the TEF will be added. It will read:

"We also estimated total exchange flow (TEF) to quantify the overall change in estuarine water renewing efficiency using the isohaline framework method proposed by MacCready (2011), which was found to be a precise way to calculate the landward transport (Chen et al., 2012). In this method, the tidally averaged volume flux of water with salinity greater than $s$ is defined as:

$$Q(s) = \left\langle \int_{A_s} u \, dA \right\rangle \quad (4)$$

where $A_s$ is the tidally varying portion of the cross-section with salinity larger than $s$. In this study, we calculated $Q(s)$ for the limited salinity bins from 0 to 35 psu with an interval of 0.5 psu. The volume flux in a specific salinity class is defined as:

$$-\frac{\partial Q}{\partial s} = -\lim_{\delta s \to 0} \frac{Q(s + \delta s/2) - Q(s - \delta s/2)}{\delta s} \quad (5)$$

where the minus sign indicates that a positive value of $-\partial Q/\partial s$ corresponds to inflow for a given salinity class. The total exchange flow ($Q_{in}$) indicating the flux of water into the estuary due to all tidal and subtidal processes, is then calculated as:

$$Q_{in} \equiv \int \frac{-\partial Q}{\partial s}\Big|_{in} ds \quad (6)$$

The resulting salt flux into the estuary ($F_{in}$) is given by:

$$F_{in} = \int s\left(-\frac{\partial Q}{\partial s}\right)\Big|_{in} ds \quad (7)$$

and the ratio of salt mass inside the estuary to the salt influx gives the mean residence time ($T_{res}$):

$$T_{res} = \frac{\int s \, dV}{F_{in}} \quad (8)$$

where $V$ is the estuarine volume."

Specific comments:

In Abstract:
1) I think the main message is to present RESULTS of the study using 3D SCHISM and not to present model (first sentence)?
We agree. The first sentence in the Abstract will be changed to "A 3D unstructured-grid hydrodynamic model was developed for the northwestern Gulf of Mexico including main estuarine systems along the Texas-Louisiana coast, with a high-resolution horizontal grid and a hybrid vertical grid."

2) If they used only Hycom boundary conditions then it is global model and not models (line 16), or if they used added tidal elevations then should state that precisely.
We will revise it as suggested. It will read "HYCOM global model"

In Methodology (2.1):
1) line 90: Does Schism use simple 1 order Galerkin method for momentum of higher order (as it does for tracers)? If not, does the authors think this is not relevant for the study where wind dynamics and momentum plays important role?

We agree that it is not relevant to mention the Galerkin method here. We will revise the sentence as "It uses highly efficient semi-implicit finite-element/finite-volume methods with a Eulerian-Lagrangian algorithm to solve the turbulence-averaged Navier-Stokes equations, including continuity, momentum, salt-balance, and heat-balance equations, under the hydrostatic approximation."

In Forcing conditions (2.3):
1) line 124: model or models?
We will revise "from the global models" to "from HYCOM global model".

2) line 142: what was used to compute heat/momentum/fresh water processes between ocean and atmosphere? If this is bulk flux then they should reference.
The model uses the bulk aerodynamic module of Zeng et al. (1998). The reference will be added. A sentence reading "The bulk aerodynamic module of Zeng et al. (1998) is used to compute the air-sea heat exchange" will be added in the model introduction (Section 2.1).

3) line 146: definition of sub-tidal period for boundary condition filtering was set to 15 days and later in the text they use 2 days? Is there particular reason why they chose 15 days and not less (i.e. 2 days) which would allow for inclusion of eddy dynamics embedded in Hycom model?
The global HYCOM doesn't provide hourly output but one instantaneous output per day. Therefore, we used a longer cut-off period to obtain sub-tidal components. The eddy condition at the open boundary was not smoothed out by this filtering process, as meso-scale eddies (e.g., loop current eddies) move slowly in the Gulf.

In Numerical experiments (2.4):
1) line 150-152: Authors used constant and the same river flux in Galveston Bay during the whole year in all 3 experiments? Did they used the same and constant fluxes for GAM in experiments Jan-GAM and July-GAM? What were the values?
Yes. We used long-term mean constant discharges into Galveston Bay for all three numerical experiments and long-term mean constant discharges from the Mississippi and Atchafalaya rivers in Jan-GAM and Jul-GAM. We will add the values in the main text of the revised manuscript (see the values in Table A of this document).

2) line 156: methodology of replicating January wind during the whole year is a bit strange; as it captures some variability within month that is replicated 12 times. What would make more sense is to use "typical winter / summer" case where they could compute multi-year mean wind field from ECMWF fields. Otherwise January/July as generic names have different meaning (authors used specific 2008 winds so they are not really generic i.e. seasonal in strict definition).
Variability in the wind is important in determining the fate of the Mississippi-Atchafalaya plume. Multi-year "mean" wind may lead to unrealistically strong stratification and weak wind mixing, as the averaging will smooth out the peak winds. The 2008 Jan and July winds are used as the wind-induced shelf currents in January and July 2008 represent typical seasonal variations in the northern Gulf. Wind roses for both months show the

dominant winds blowing from distinctively different directions, with the January wind mainly blowing from NE, E, and SE, while the July wind blowing mainly from S. Such distinctively different wind patterns cause great differences in shelf transport (Fig. D) and thus the distribution of low-salinity water from the Mississippi-Atchafalaya rivers.

[Figure]

Figure C: Wind roses for January and July of 2008 at the Galveston Bay mouth. (Note: this figure will be put in supplemental material in the revised manuscript).

[Figure]

Figure D: (a) The location of four selected cross-shelf sections on the Texas-Louisiana shelf and (b) the downcoast shelf transport for three numerical experiments. The grey lines in (a) indicate the 50, 100, 150, and 200 m bathymetric contours. (Note: this figure will be added in the revised manuscript).

3) line 160: I am not sure what authors mean with computing boundary conditions form 2 years temporally constant?
To make the wind field the only controlling factor in the numerical experiments, we used the realistic boundary conditions in the 2007-2008 model run for all other variables. The sentence in the text is mistaken. It will read in the revised manuscript as "Except for the wind forcing, river input, and initial salinity condition, the numerical experiments use the same model configuration as in the realistic 2007-2008 model run."

In Water level (3.1):
1) in line 180: Why not to state what is the Cd equivalent to Manning coef as authors used quadratic bottom friction, instead of reporting Manning's coef? What was the method and how they tuned Manning coef is not really clear.
The drag coefficient, calculated in the model as a function of the Manning coefficient and total water depth, varies spatially and temporally. It is therefore not feasible to provide an equivalent value of the drag coefficient for the given Manning coefficient.

For the model calibration, we carried out multiple model runs with different Manning coefficient ranging from 0.015 to 0.025 and chose 0.016 as it gives the best reproduction of the tidal amplitude and phase. Figure E shows how the harmonic water level is reproduced by the model with the Manning coefficient of 0.016.

[Figure]

Figure E: Harmonic water level comparison between model and observation at four selected stations: Bob Hall (Texas), Freeport (Texas), Pilot Station (Louisiana), and Dauphin Island (Alabama). (Note this figure will be added in the supplemental material).

2) line 185: They speculate that low skill at Pilot Station is due to proximity of boundary conditions, which seems not plausibly as boundary conditions are fare away. Another point is that low-frequency MAE (i.e. boundary effect contribution) is much better than total which implies that other dynamics is important contribution to the MAE (Table 1)? Thanks for your careful reading. We double checked the model-observation statistics and found an error for the Pilot Station. For the total water level, the *MAE* and skill are 7 cm and 0.93; we will update the table. Pilot Station and Dauphin Island show the poorest skills for the subtidal water level. The revised manuscript will have "The *MAE* is in the range of 7-9 cm and 5-7 cm for the total and subtidal components, respectively. The model skill varies spatially, with relatively low skills for the subtidal components at Pilot Station and Dauphin Island."

In Salinity (3.2):
line 204: How authors explain lower MAE for global than low-passed filtered case in BOLI station? This seems hard to believe in mathematical sense, possibly some error. We double checked the model-observation statistics and found an error for the *MAE* at BOLI. The *MAE* is 4 psu for total salinity and 4 psu for subtidal; we will update the table. We also checked all other statistics in the table (thanks for your careful reading).

In 4. (Remote influence):
line 263-264: Sentence is not clear and make no sense: "Horizontally, their distribution influences but is also regulated by the shelf dynamics, and exhibits significant seasonal variation."
To make the sentence clearer, we will revise it as "Horizontally, their distribution is regulated greatly by the shelf dynamics and thus usually exhibits significant seasonal variation due to seasonal variation of wind and shelf circulation."

lines 336-351 should include equation how they computed residence time.
Equations for the salt flux decomposition, the TEF, and the residence time will be added in the revised manuscript.

In 5. (Summary): I think that authors should emphasize main results from their study and answers they provided on questions posed in the last paragraph of introduction (i.e. time needed for information originating at Mississippi-Atch rivers to arrive to GB? About extent and portion of seasonal influence of winds to the horizontal distribution of salinity etc.). This way written summary seems too short and doesn't summaries the study. It is a good suggestion to summarize what we found from the numerical experiments. We also plan to add additional analysis for the mixing and shelf transport under the influence of Mississippi-Atchafalaya discharge and different wind conditions (see Fig. F).

We will add the following paragraph (as the 2$^{nd}$ paragraph) to Summary:
"Three numerical experiments were carried out to examine the extent to which the neighboring major rivers influence a local coastal system. The Mississippi-Atchafalaya discharge has great influence on the salinity regime along the Texas coast and its influence depends on the wind forcing and the resulting shelf circulation. Winter wind causes a stronger downcoast shelf transport, an order of magnitude larger than that during

summer (Fig. D), transporting the Mississippi-Atchafalaya plume to the Texas coast. The mean discharge from the Mississippi-Atchafalaya rivers can lower the salinity by up to 10 psu at the mouth of Galveston Bay under winter wind. Lower salinity condition on the Texas shelf decreases the longitudinal salinity gradient inside the estuary, leading to a weakened estuarine circulation and weaker salt exchange. The vertical mixing is also sensitive to the wind forcing. The low-salinity water expands further offshore with summer wind, while it is constrained as a narrow ban against the coastline with winter wind. As a result, stratification is stronger over the shelf, inhibiting the vertical mixing on the shelf, during summer."

[Figure]

Figure F: Salinity (left panels) and vertical diffusivity (right panels) averaged over days 250-300 for the section through Trinity Bay, ship channel, and Texas shelf: see the inset in (a) for the section location. (Note: this figure will be added in the revised manuscript).

---

## Referee Comment (RC3) · Anonymous Referee #1 · 18 Apr 2019

The authors addressed my questions very well. I do not have any other problems.

---

## Referee Comment (RC4) · Anonymous Referee #2 · 24 Apr 2019

Authors responded to all my comments and improved the manuscript at the level that is acceptable for publication.

---

## Author Comment (AC3) · 30 Apr 2019

Thanks again for your encouragement and helpful comments.

———————————————

---

## Author Comment (AC4) · 30 Apr 2019

Thanks again for your helpful comments. Your suggestions about adding more discussion on the mixing and stratification have improved the manuscript a lot.
* * *

---

## Author Response (AR1)

General response
We want to express our sincere thanks to both reviewers for their helpful comments and suggestions. We are glad to see both reviewers are satisfied our responses during the interactive discussion. The new version of manuscript is modified in the way we have planned during the discussion (therefore, there will be little difference between the following responses and the ones we have uploaded during the interactive discussion). For your convenience, we summarized the major modifications here.
1. New figures (Fig. 11 & 14) are added to examine the influence of Mississippi-Atchafalaya Rivers' influence on the stratification, mixing, and longshore transport. Corresponding text is added in Section 4.2 and 4.3.
2. Three supplemental figures are added. They are used to show the wind rose for 2008 January and July, tidal signal comparison for water level between model and observation, and shelf circulation for the three numerical experiments.
3. Index used to demonstrate the agreement between model and observation are updated in Table 1.
4. Almost all the figures are replotted in order to make the plot style consistent (e.g., tick direction inward instead of outward).
5. A new paragraph summarizing the paper is added in the conclusion Section.

**Responses to comments by reviewer #1**

The manuscript "A hydrodynamic model for Galveston Bay and the shelf in the northwestern Gulf of Mexico" by Du et al. presents a detailed study on development, validation and implementation of an unstructured grid model in the Gulf of Mexico and adjacent estuarine bays. The impact of the remote river discharge from Mississippi and Atchafalaya on the hydrodynamics in the Galveston Bay is assessed. Overall, the model was well established, and carefully validated. The model results are quite reliable. A large-scale unstructured model is very helpful in identifying the effect of shelf processes on a specific estuary. The unstructured model provides great flexibility in covering the large GOM domain and resolving the complex shoreline and bathymetry in the estuaries. This study presents a good example of research on estuary-shelf interaction. The manuscript is well written and organized, and several new findings are provided, thus it merits a publication after some minor revision.

Thanks very much for the helpful comments and suggestions.

My main suggestions are:
1) More focus could be paid on the physics, like how the buoyancy induced coastal current from the Atchafalaya and Mississippi Rivers affects the advective transport and mixing at the mouth of the Galveston Bay;

It is a good suggestion. In the revised manuscript, we add Figure A (Fig. 14 in MS) showing the salinity and vertical diffusivity profile along a section through Trinity Bay, ship channel, and Texas Shelf off Galveston Bay. The stratification, salt intrusion, as well

as the vertical mixing, is further discussed in Section 4.3. Some interesting points include: 1) salt intrusion is enhanced by July wind; 2) July wind also causes a stronger stratification on the Texas shelf off Galveston Bay; and 3) vertical mixing is reduced in the area where salinity-induced stratification is enhanced, such as the shelf region and near the bay entrance.

[Figure]

Figure A: Salinity (left panels) and logarithm of vertical diffusivity (right panels) for the section through Trinity Bay, ship channel, and Texas Shelf: see the inset in (a) for the section location.

2) How about the advantage of this unstructured grid in resolving the narrow and deep channel? How important is this resolution in reproducing salt intrusion? Based on my previous experience, this model seems a little difficult to simulate salt intrusion in the Pearl River Estuary. The model results always seem overmixed than observations;

This is a good point worthy of discussion. The necessity of resolving the deep ship channel has already been investigated by Rayson et al. (2015-Ocean Modeling). This study compared three different grids (with/without resolving the ship channel) in Galveston Bay, and demonstrated that salinity and sub-tidal velocity are better predicted when the important topographic features (e.g., ship channel) are well resolved. It is well known that the deep channel plays a key role in advecting saltwater from coastal ocean to an estuary. We did not have measurement showing the salt intrusion along the channel, but comparison with the salinity observations at the mid-bay station clearly shows that our model can reproduce the salinity variation at this station, which cannot be achieved without reproducing the salt intrusion processes.

For the over-mixed problem in some estuarine applications, here are some suggestions or experiences we have.

(1) Vertical grid matters a lot. There are several options in the SCHISM for the vertical grid, including pure sigma grid, hybrid s-z grid, and more advanced LSC$^2$. It would be important to have fine resolution at the pycnocline.
(2) Bathymetry and horizontal grid are important to resolve sharply-changing bathymetric features, such as the ship channel. The SCHISM model doesn't require bathymetry smoothing. Bathymetry smoothing may induce some unrealistic vertical mixing (see Ye et al., 2018-Ocean Modeling).
(3) The boundary condition might play a role. Both river input and the stratification at the ocean boundary can affect the stratification. The ocean boundary condition matters especially when the target estuary is not that far away from the ocean boundary.

3) How Galveston and other estuaries affect the shelf processes, e.g., how they impact the coastal current, and water column stratification and mixing?

It is a good suggestion to further discuss the effect of river input on the shelf processes. We add another figure to show how the shelf transport is affected by the Mississippi-Atchafalaya River input as well as the wind forcing (Fig. B, or Fig. 11 in MS), which shows that the Mississippi-Atchafalaya River input enhances the shelf transport by 10-14% and that the July wind dramatically reduces the downcoast shelf transport. We also put additional discussion on stratification and mixing based on Fig. A, which shows interesting changes in terms of water column stratification and mixing. One of the most obvious change is the stronger stratification and weaker vertical mixing over the Texas Shelf under July wind, compared to that under January wind.

[Figure]

Figure B: (a) The location of four selected sections over the Texas-Louisiana Shelf and (b) the downcoast shelf transport for three numerical experiments. The grey lines in (a) indicate the 50, 100, 150, and 200 m bathymetric contours.

**Response to comments by reviewer #2**

Manuscript entitled "A hydrodynamic model for Galveston Bay and the shelf in the northwestern Gulf of Mexico" by Jiabi Du et al. presents a study of the influences and effects that seasonal wind forcing has on the salinity distribution along the Louisiana shelf with focus on Galveston Bay. As this problem is having multi-scale dependency, i.e. shelf dynamics has important influence on the coastal dynamics and vice versa, unstructured modelling approach seems to give reasonable answers and is appropriate. Manuscript is well organized, with some additional information should be valid contribution and appropriate for the journal.

Thanks very much for your comments, which have helped clarify multiple potentially confusing statements in the manuscript.

General comments:

1) Study is covering big portion of work done, however I think readers would benefit from clear and possibly additional explanation of baseline method used in the study. In other words, it is not clear to me if authors assumed and explored exclusively wind (January vs. July) effects in the 3 numerical simulations (Jan-G, Jan-GAM, July-GAM) using the same wind field (Jan in first 2 or July in 3rd) replicated in time during the whole year without using any heat flux (or other atmospheric model forcing) or tidal forcing. One table listing used/or not used assumptions would help (i.e. no heat flux, no boundary conditions from Hycom, no tides, initial field, winds from Jan or July replicated during the whole simulation).

The numerical experiments have virtually the same setting as in the realistic 2007-2008 model run, including tide, heat exchange, and ocean boundary conditions. The only differences from the realistic run are: (1) an initial salinity condition of 36 psu over the entire domain; (2) using repeated monthly wind forcing of January or July 2008; (3) long-term mean constant river discharges from three estuarine systems (Mississippi, Atchafalaya, and Galveston Bay). The table below shows the differences in numerical experiments relative to the realistic run. With only three differences, we do not plan to add this table in the revised manuscript. Instead, we add additional sentences describing the setting of the numerical experiments in the revised manuscript (L160-176).

Table A: Settings for the numerical experiments.

|  | Jan-G | Jan-GAM | Jul-GAM |
|---|---|---|---|
| River discharge (m3/s) | Q(Trinity River)=190
Q(San Jacinto)=52
Q(Buffalo Bayou)=14
Q(Chocolate Bayou)=3 | Q in Jan-G, plus
Q(Atchafalaya River)=15525
Q (Mississippi River)=6664 | |
| Wind | Repeated wind of January 2008 | | Repeated wind of July 2008 |
| Initial condition for salinity | 36 psu over the entire domain throughout the water column | | |
| Initial condition for temperature | Same as in realistic model run, based on HYCOM output on 2007-01-01 | | |

| Air temperature, pressure, solar radiation | Same as in realistic model run, spatially and temporally varying from 2007-01-01 |
| --- | --- |
| Ocean boundary condition | Same as in realistic model run, spatially and temporally varying from 2007-01-01 |

2) If this is the case (when there is no heat flux, but initial stratification), then the simulation represent only partly barotropic approach which is valid in shallow part of the domain and during the winter only - as there is no vertical heat flux supporting vertical stratification in balance with vertical mixing parametrized with turbulence. I would be surprised that model didn't vertically mixed the whole vertical column as 1 year of simulation is quite enough time. In the case they used vertically uniform density field for start then I have doubts it does represent valid approximation of GOM in July. Possibly, then more correct title of the experiments would be to call (and explain) those sensitivity experiments as sensitivity of the salinity field to the wind field effects (and not mix that with July/January as seasons) of the simple barotropic system interacting with coastline and bathymetry. In that case dynamics will be only due to buoyancy effects of the rivers via salinity and some wind mixing/transport without any temperature variations. Validation period using full forcing is then confirmation of model setup and tunning.

The heat flux is included in numerical experiments, same as in the realistic 2007-2008 model run. We clarify this in the revised manuscript (L160-162). The temperature profile at a station in the deep Gulf (Fig. C) indicates that in this baroclinic model run, the deep ocean is persistently stratified, with the surface temperature changing seasonally. The stratification at the bay mouth also persists (Fig. D).

[Figure]

Figure C: The temperature profile (low-pass filtered with a cut-off period of 50 h) at a station in the deep Gulf (see the inset for the station location): the grey lines in the inset denoting the 50, 100, 150, and 200 m bathymetric contours. (Note: we do not include this figure in the revised manuscript).

[Figure]

Figure D: (a) Surface salinity at the mouth of Galveston Bay (the bold lines indicating the sub-tidal salinity) and (b-d) the vertical salinity profiles at the mouth (low-passed filtered with a cut-off period of 50 h). The periodicity in (b-d) is the tropic-equatorial cycle. Note: we do not include this figure (showing the water column is not well mixed in this a full 3D baroclinic mode run) in the revised manuscript.

3) Part with residence time (336-351) seems as added to the manuscript without needed explanation of method MacCready (2011). Authors could at least give basic equations for completeness of the study and to show how they computed values.

The equations for the salt flux including the salt flux decomposition and the TEF is added (L364-382). It reads:

"We also estimated total exchange flow (TEF) to quantify the overall change in estuarine water renewing efficiency using the isohaline framework method proposed by MacCready (2011), which was found to be a precise way to calculate the landward transport (Chen et al., 2012). In this method, the tidally averaged volume flux of water with salinity greater than $s$ is defined as:

$$Q(s) = \left\langle \int_{A_s} u \, dA \right\rangle \quad (4)$$

where $A_s$ is the tidally varying portion of the cross-section with salinity larger than $s$. In this study, we calculated $Q(s)$ for the limited salinity bins from 0 to 35 psu with an interval of 0.5 psu. The volume flux in a specific salinity class is defined as:

$$-\frac{\partial Q}{\partial s} = -\lim_{\delta s \to 0} \frac{Q(s + \delta s / 2) - Q(s - \delta s / 2)}{\delta s} \quad (5)$$

where the minus sign indicates that a positive value of $-\partial Q/\partial s$ corresponds to inflow for a given salinity class. The total exchange flow ($Q_{in}$) indicating the flux of water into the estuary due to all tidal and subtidal processes, is then calculated as:

$$Q_{in} \equiv \int \frac{-\partial Q}{\partial s}\Big|_{in} ds \quad (6)$$

The resulting salt flux into the estuary ($F_{in}$) is given by:

$$F_{in} = \int s\left(-\frac{\partial Q}{\partial s}\right)\Big|_{in} ds \quad (7)$$

and the ratio of salt mass inside the estuary to the salt influx gives the mean residence time ($T_{res}$):

$$T_{res} = \frac{\int s \, dV}{F_{in}} \quad (8)$$

where $V$ is the estuarine volume."

Specific comments:

In Abstract:
1) I think the main message is to present RESULTS of the study using 3D SCHISM and not to present model (first sentence)?
We agree. The first sentence in the Abstract will be changed to "A 3D unstructured-grid hydrodynamic model for the northern Gulf of Mexico was developed, with a hybrid s-z vertical grid and high-resolution horizontal grid for the main estuarine systems along the Texas-Louisiana coast. This model, based on the Semi-implicit Cross-scale Hydroscience Integrated System Model (SCHISM), is driven by the observed river discharge, reanalysis atmospheric forcing, and open boundary conditions from global HYCOM output."

2) If they used only Hycom boundary conditions then it is global model and not models (line 16), or if they used added tidal elevations then should state that precisely.
Revised as suggested. It now read "global HYCOM output"

In Methodology (2.1):

1) line 90: Does Schism use simple 1 order Galerkin method for momentum of higher order (as it does for tracers)? If not, does the authors think this is not relevant for the study where wind dynamics and momentum plays important role?
We agree that it is not relevant to mention the Galerkin method here. We will revise the sentence as "It uses highly efficient semi-implicit finite-element/finite-volume methods with a Eulerian-Lagrangian algorithm to solve the turbulence-averaged Navier-Stokes equations, including continuity, momentum, salt-balance, and heat-balance equations, under the hydrostatic approximation."

In Forcing conditions (2.3):
1) line 124: model or models?
We will revise "from the global models" to "from global HYCOM output".

2) line 142: what was used to compute heat/momentum/fresh water processes between ocean and atmosphere? If this is bulk flux then they should reference.
The model uses the bulk aerodynamic module of Zeng et al. (1998). The reference will be added. A sentence reading "The bulk aerodynamic module of Zeng et al. (1998) is used to compute the air-sea heat exchange" is added (L152-153).

 3) line 146: definition of sub-tidal period for boundary condition filtering was set to 15 days and later in the text they use 2 days? Is there particular reason why they chose 15 days and not less (i.e. 2 days) which would allow for inclusion of eddy dynamics embedded in Hycom model?
The global HYCOM doesn't provide hourly output but one instantaneous output per day. Therefore, we used a longer cut-off period to obtain sub-tidal components. The eddy condition at the open boundary was not smoothed out by this filtering process, as meso-scale eddies (e.g., loop current eddies) move slowly in the Gulf.

In Numerical experiments (2.4):
1) line 150-152: Authors used constant and the same river flux in Galveston Bay during the whole year in all 3 experiments? Did they used the same and constant fluxes for GAM in experiments Jan-GAM and July-GAM? What were the values?
Yes. We used long-term mean constant discharges into Galveston Bay for all three numerical experiments and long-term mean constant discharges from the Mississippi and Atchafalaya rivers in Jan-GAM and Jul-GAM. We add the values in the main text of the revised manuscript (see the values in Table A of this document; L171-175).

2) line 156: methodology of replicating January wind during the whole year is a bit strange; as it captures some variability within month that is replicated 12 times. What would make more sense is to use "typical winter / summer" case where they could compute multi-year mean wind field from ECMWF fields. Otherwise January/July as generic names have different meaning (authors used specific 2008 winds so they are not really generic i.e. seasonal in strict definition).
Variability in the wind is important in determining the fate of the Mississippi-Atchafalaya plume. Multi-year "mean" wind may lead to unrealistically strong stratification and weak wind mixing, as the averaging will smooth out the peak winds. The 2008 Jan and July

winds are used as the wind-induced shelf currents in January and July 2008 represent typical seasonal variations in the northern Gulf. Wind roses for both months show the dominant winds blowing from distinctively different directions, with the January wind mainly blowing from NE, E, and SE, while the July wind blowing mainly from S (Fig. E). Such distinctively different wind patterns cause great differences in shelf transport (Fig. F) and thus the distribution of low-salinity water from the Mississippi-Atchafalaya rivers.

[Figure]

Figure E: Wind roses for January and July of 2008 at the Galveston Bay mouth. (Note: this figure is put in supplemental material in the revised manuscript).

3) line 160: I am not sure what authors mean with computing boundary conditions form 2 years temporally constant?
To make the wind field the only controlling factor in the numerical experiments, we used the realistic boundary conditions in the 2007-2008 model run for all other variables. The sentence in the text is mistaken. It will read in the revised manuscript as "To investigate the remote influence from the MAR discharge, we conducted three numerical experiments that use the same model configuration as in the realistic 2007-2008 model run except for freshwater discharge, wind forcing, and initial salinity conditions."

In Water level (3.1):
1) in line 180: Why not to state what is the Cd equivalent to Manning coef as authors used quadratic bottom friction, instead of reporting Manning's coef? What was the method and how they tuned Manning coef is not really clear.
The drag coefficient, calculated in the model as a function of the Manning coefficient and total water depth, varies spatially and temporally. It is therefore not feasible to provide an equivalent value of the drag coefficient for the given Manning coefficient.

For the model calibration, we carried out multiple model runs with different Manning coefficient ranging from 0.015 to 0.025 and chose 0.016 as it gives the best reproduction of the tidal amplitude and phase. Figure F shows how the harmonic water level is reproduced by the model with the Manning coefficient of 0.016.

[Figure]

Figure F: Harmonic water level comparison between model and observation at four selected stations: Bob Hall (Texas), Freeport (Texas), Pilot Station (Louisiana), and Dauphin Island (Alabama). (Note this figure is added in the supplemental material).

2) line 185: They speculate that low skill at Pilot Station is due to proximity of boundary conditions, which seems not plausibly as boundary conditions are fare away. Another point is that low-frequency MAE (i.e. boundary effect contribution) is much better than total which implies that other dynamics is important contribution to the MAE (Table 1)? Thanks for your careful reading. We double checked the model-observation statistics and found an error for the Pilot Station. For the total water level, the *MAE* and skill are 7 cm and 0.93; we update the number in Table 1. Pilot Station and Dauphin Island show the poorest skills for the subtidal water level. The revised manuscript will have "The MAE is in the range of 7-8 cm and 5-7 cm for the total and subtidal components, respectively. The model skill varies spatially, with relatively low skills (0.88) at Pilot Station and Dauphin Island for the subtidal component and high skills ($\geqslant$ 0.94) at the stations in the Texas coast including Galveston Bay for both the total and subtidal components."

In Salinity (3.2):
line 204: How authors explain lower MAE for global than low-passed filtered case in BOLI station? This seems hard to believe in mathematical sense, possibly some error. We double checked the model-observation statistics and found an error for the *MAE* at BOLI. The *MAE* is 4 psu for total salinity and 4 psu for subtidal; we have updated the table. We also checked all other statistics in the table (thanks for your careful reading).

In 4. (Remote influence):
line 263-264: Sentence is not clear and make no sense: "Horizontally, their distribution influences but is also regulated by the shelf dynamics, and exhibits significant seasonal variation."
We remove the entire paragraph as the focus of the paragraph is about the river plumes, which is not well suited with the reorganized section 4.1-4.4. Instead, we replace the

paragraph with "The conditions in Texas coastal waters are impacted by several remote sources, including mesoscale eddies (Oey et al., 2005; Ohlmann and Niiler, 2005), longshore transport of low-salinity water from major rivers (Li et al., 1997; Nowlin et al., 2005), and Ekman transport induced by longshore wind and the resulting upwelling/downwelling (Li et al., 1997; Zhang et al., 2012). Here, based on the realistic model results and numerical experiments, we discuss the remote influence of major river plumes and shelf dynamics on the longshore transport, salinity, stratification, and vertical mixing at the Texas coast, as well as the water exchange between the coastal ocean and local coastal system."

lines 336-351 should include equation how they computed residence time.
Equations for the salt flux decomposition, the TEF, and the residence time are added in the revised manuscript.

In 5. (Summary): I think that authors should emphasize main results from their study and answers they provided on questions posed in the last paragraph of introduction (i.e. time needed for information originating at Mississippi-Atch rivers to arrive to GB? About extent and portion of seasonal influence of winds to the horizontal distribution of salinity etc.). This way written summary seems too short and doesn't summaries the study.
It is a good suggestion to summarize what we found from the numerical experiments.

A new paragraph (as the 2nd paragraph) is added to Summary:

[revised manuscript text omitted]